# Emergence of non-canonical parvalbumin-containing interneurons in hippocampus of a murine model of type I lissencephaly

**Tyler G Ekins[1,2], Vivek Mahadevan[1], Yajun Zhang[1], James A D'Amour[1,3], Gülcan Akgül[1], Timothy J Petros[1], Chris J McBain[1]\***

[1]Program in Developmental Neurobiology, Eunice Kennedy-Shriver National Institute of Child Health and Human Development, National Institutes of Health, Bethesda, United States; [2]NIH-Brown University Graduate Partnership Program, Providence, United States; [3]Postdoctoral Research Associate Training Program, National Institute of General Medical Sciences, Bethesda, United States

**Abstract** Type I lissencephaly is a neuronal migration disorder caused by haploinsuffiency of the *PAFAH1B1* (mouse: *Pafah1b1*) gene and is characterized by brain malformation, developmental delays, and epilepsy. Here, we investigate the impact of *Pafah1b1* mutation on the cellular migration, morphophysiology, microcircuitry, and transcriptomics of mouse hippocampal CA1 parvalbumin-containing inhibitory interneurons (PV+INTs). We find that WT PV+INTs consist of two physiological subtypes (80% fast-spiking (FS), 20% non-fast-spiking (NFS)) and four morphological subtypes. We find that cell-autonomous mutations within interneurons disrupts morphophysiological development of PV+INTs and results in the emergence of a non-canonical 'intermediate spiking (IS)' subset of PV+INTs. We also find that now dominant IS/NFS cells are prone to entering depolarization block, causing them to temporarily lose the ability to initiate action potentials and control network excitation, potentially promoting seizures. Finally, single-cell nuclear RNAsequencing of PV+INTs revealed several misregulated genes related to morphogenesis, cellular excitability, and synapse formation.

**\*For correspondence:**
mcbainc@mail.nih.gov

**Competing interests:** The authors declare that no competing interests exist.

## Introduction

Excitation in neocortical and hippocampal circuits is balanced by a relatively small (10–15%) yet highly heterogenous population of GABAergic inhibitory interneurons (*Pelkey et al., 2017*). During embryogenesis inhibitory interneurons are generated in the ganglionic eminences, then tangentially migrate to reach their cortical or hippocampal destination (*Bartolini et al., 2013*). Upon reaching the cortex or hippocampus, interneurons migrate along radial glial cells to their final laminar position and integrate into developing circuits (*Lim et al., 2018*). The process of neuronal migration requires molecular interactions of motor proteins with networks of microtubules; therefore, genetic mutations disrupting such proteins compromise neuronal migration and promote abnormal brain development (*Wynshaw-Boris, 2001*; *Corbo et al., 2002*; *Tissir and Goffinet, 2003*).

*Pafah1b1* encodes a protein (Pafah1b1) that regulates dynein microtubule binding and is essential for neuronal migration (*Wynshaw-Boris, 2001*). Consequently, *Pafah1b1* haploinsufficiency results in classical, or Type I, lissencephaly ('smooth brain'), a rare neurodevelopmental disorder characterized in humans by brain malformation, intellectual disability, motor impairment, and drug-resistant epilepsy (*Kato, 2003*; *Di Donato et al., 2017*). Full loss of *Pafah1b1* is embryonically lethal (*Hirotsune et al., 1998*).

Classical lissencephaly can be modeled in mouse lines generated through heterozygous removal of *Pafah1b1*, which results in enlarged ventricles and disorganization of brain structures

(*Hirotsune et al., 1998*). Structural abnormalities are particularly notable in the hippocampus, where the normally tightly compacted layer of pyramidal cells fractures into multiple bands of loosely organized cells (*Fleck et al., 2000*; *D'Amour et al., 2020*). Mice heterozygous for *Pafah1b1* share symptoms with human lissencephaly patients, including learning deficits, motor impairments, increased excitability and decreased seizure threshold (*Paylor et al., 1999*; *Fleck et al., 2000*; *Greenwood et al., 2009*; *Menascu et al., 2013*; *Herbst et al., 2016*). Due to the high density of recurrent excitatory connections and the reliance on inhibitory interneurons to control network excitability, the hippocampus and neocortex are prone to generating epileptic seizures (*McCormick and Contreras, 2001*). Thus, the increased propensity for seizures in *Pafah1b1* mutants may be indicative of dysfunctional inhibition. Indeed, specific deficits in inhibitory interneuron wiring with pyramidal cell targets have been identified in *Pafah1b1* mutant mice, but the origin of seizures remains unclear (*Jones and Baraban, 2009*; *D'Amour et al., 2020*).

Inhibitory interneurons are classified based on a combination of their morphological, biochemical, intrinsic electrical, and connectivity properties (*Lim et al., 2018*). Advances in single-cell RNA sequencing have revealed enormous diversity in interneuron genomics, and current efforts attempt to correlate transcriptomic data sets with previously identified interneuron subtypes (*Tasic et al., 2018*; *Muñoz-Manchado et al., 2018*; *Gouwens et al., 2019*; *Que and Lukacsovich, 2020*). In CA1 hippocampus alone, inhibitory synaptic transmission is mediated by at least 15 different subtypes of GABAergic inhibitory interneurons (*Pelkey et al., 2017*). Three canonical interneuron subtypes express the calcium-binding protein parvalbumin (PV): basket-cells, axo-axonic cells, and bistratified cells. PV-containing inhibitory interneurons (PV+INTs) are often classified as 'fast-spiking' cells due to their ability to sustain high-frequency discharges of action potentials with minimal spike-frequency adaptation/accommodation (*Pelkey et al., 2017*). Fast-spiking interneurons are essential for proper network oscillations and disrupting the function of PV+INTs can generate spontaneous recurrent seizures (*Drexel et al., 2017*; *Panthi and Leitch, 2019*). Recent transcriptomics suggests that there are several genomically distinct subpopulations of PV+INTs (*Hodge et al., 2019*; *Gouwens et al., 2020*), some of which may correspond to unique PV+INT subtypes that have remained largely understudied relative to the canonical FS subtypes listed above.

A current model for the formation of neural circuits posits that pyramidal cells (PCs) instruct radial migration and synaptic connectivity of INTs (*Pelkey et al., 2017*; *Wester et al., 2019*). In the cortex, INTs are initially dispersed throughout cortical layers, only sorting into their final positions between the 3rd and 7th postnatal day (*Miyoshi and Fishell, 2011*). Interneurons have programs that enable both cell-type-specific and cellular compartment-specific targeting. For example, PV+INTs make connections with PCs and other PV+INTs, but rarely contact other subtypes of INTs (*Kohus et al., 2016*). Furthermore, different subtypes of PV+INTs target-specific regions of PCs such as dendrites (bistratified cells), the axon initial segment (axo-axonic cells), or the perisomatic region (basket cells; *Pelkey et al., 2017*). Mutations to chemokine receptors can alter this connectivity, and complete loss or reprogramming of cellular identity is possible when proteins are missing in development (*Ye et al., 2015*; *Pelkey et al., 2017*; *Mahadevan and Mitra, 2020*).

Previous lissencephaly studies have demonstrated that migration of inhibitory interneurons is disrupted in *Pafah1b1* heterozygous mutants (*Fleck et al., 2000*; *McManus et al., 2004*). In particular, PV+INTs adopt atypical positions in the hippocampus, including between heterotopic bands of pyramidal cells and within stratum radiatum, a layer where PV+INTs are rarely found in wildtype (WT) CA1 (*Fleck et al., 2000*; *Jones and Baraban, 2009*; *D'Amour et al., 2020*). Despite ectopic positioning of inhibitory interneurons and layer-specific reorganization of inhibitory inputs, the nature and consequences of PV+INT morphophysiological development and microcircuit organization following *Pafah1b1* mutations have remained elusive.

Here, we report the impact of *Pafah1b1* mutations and resulting neuronal migration deficits on the lamination, morphology, intrinsic physiology, connectivity, synaptic transmission dynamics, and genomics of hippocampal parvalbumin-containing inhibitory interneurons. Cell-autonomous loss of *Pafah1b1* within interneurons results in the emergence of a novel physiological population of PV+INT, comprising ~50% of the total PV+INT cohort. Compared to canonical FS PV+INTs, these altered cells have lower firing rates, provide less reliable inhibition to pyramidal cells and have a higher propensity to enter depolarization block. Single-cell nuclear RNA sequencing (snRNA-seq) revealed multiple disruptions to the expression of ion channels regulating PV+INT excitability. We

propose that disrupted physiological development and deficient inhibitory output of PV+INTs likely contributes to the spontaneous seizures observed in classical lissencephaly.

## Results and discussion

### Generation and characterization of *Pafah1b1* mutant lines

To investigate the cell-autonomous and non-autonomous effects of *Pafah1b1* heterozygous mutations on PV+INT migration and development, we crossed *Pafah1b1*^floxedl/+ breeders to three separate Cre lines: Sox2-Cre to generate heterozygous *Pafah1b1* mutations in all cells ('GlobalLis'); Nkx2.1-Cre to generate heterozygous *Pafah1b1* mutations specifically in medial ganglionic eminence-derived interneurons ('NkxLis'); and Emx1-Cre to generate heterozygous mutations specifically in pyramidal cells ('EmxLis'). These lines were further crossed to PV-TdTomato (TdT) reporter lines to enable selective targeting of PV+INTs during physiological recordings.

As previously reported (*Hirotsune et al., 1998*; *Fleck et al., 2000*; *D'Amour et al., 2020*) hippocampal lamination is disrupted in GlobalLis mice. The normally compact layer of pyramidal cells (PCs) (stratum pyramidale; s.p.) fractures into heterotopic bands, typically with a normotopic layer resembling the WT band, and an ectopic layer often fragmented into stratum oriens (s.o.; *Figure 1A*). This general pattern of disrupted hippocampal lamination is also observed in the EmxLis mouse line, but not in the NkxLis mouse, indicating that *Pafah1b1* expression in pyramidal cells, but not MGE-derived interneurons, is essential for proper hippocampal pyramidal cell layer formation (*Figure 1A*).

### Radial migration of inhibitory interneurons is disrupted in *Pafah1b1* mutants

Previous studies have demonstrated severe cellular disorganization in rodent models of lissencephaly, including impaired radial migration of parvalbumin-containing inhibitory interneurons (PV+INTs; *Fleck et al., 2000*; *Jones and Baraban, 2009*; *D'Amour et al., 2020*). To investigate the nature of the aberrant migration of PV+INTs, we first quantified their relative densities using immunohistochemistry. GlobalLis mutants exhibited no overall change in PV density in the CA1 subfield. However, as we reported previously (*D'Amour et al., 2020*) PV+INT density was reduced in both s.o. and s.p. and increased in stratum radiatum (s.r.) and stratum lacunosum-moleculare (s.l.m.; *Figure 1B–C*). In WT CA1, an overwhelming majority (>95%) of PV+INTs are found in s.o. and s.p., while less than 5% reside in s.r. or s.l.m., indicating a strong preference for PV+INTs to typically inhabit deeper regions of the hippocampus. In contrast, in GlobalLis 77% of PV+INTs were found in s.o./s.p, with the proportion of cells in s.r./s.l.m. expanded to 23% of the total PV population (*Figure 1D*).

We next used the NkxLis and EmxLis lines, where *Pafah1b1* is eliminated only in MGE-derived interneurons and pyramidal neurons respectively, to assay the impact of cell-autonomous and non-autonomous mutations on PV+INT migration. Interestingly, both genotypes had similar patterns of PV+INT somatic distribution to that observed in the GlobalLis CA1. In these mutants, there was a significantly decreased density of PV+INTs in s.o./s.p. and increased density in s.r./s.l.m., with 80 (NkxLis)−82 (EmxLis) % of PV+INTs found in deep regions (s.o/s.p.) and 18 (EmxLis)−20 (NkxLis) % found in superficial regions (s.r./s.l.m.; *Figure 1B–D*). Thus, proper migration and lamination of PV+INTs requires both *Pafah1b1*-dependent cell-intrinsic mechanisms (as revealed by the disruption in NkxLis PV cells) and non-cell-autonomous cues from pyramidal neurons (demonstrated by the disruption in EmxLis PV cells).

### *Pafah1b1* heterozygous mutation alters PV+INT morphophysiological development

The vast majority of parvalbumin-containing inhibitory interneurons are classified as 'fast-spiking' (FS) cells, due to their ability to sustain high-frequency discharges of action potentials (*Pelkey et al., 2017*). In cortical circuits, FS cells contribute to both feedforward and feedback inhibition and are essential in generating network oscillations; as such, disrupted FS cell function can lead to uncontrolled excitation and seizures (*Hu et al., 2014*).

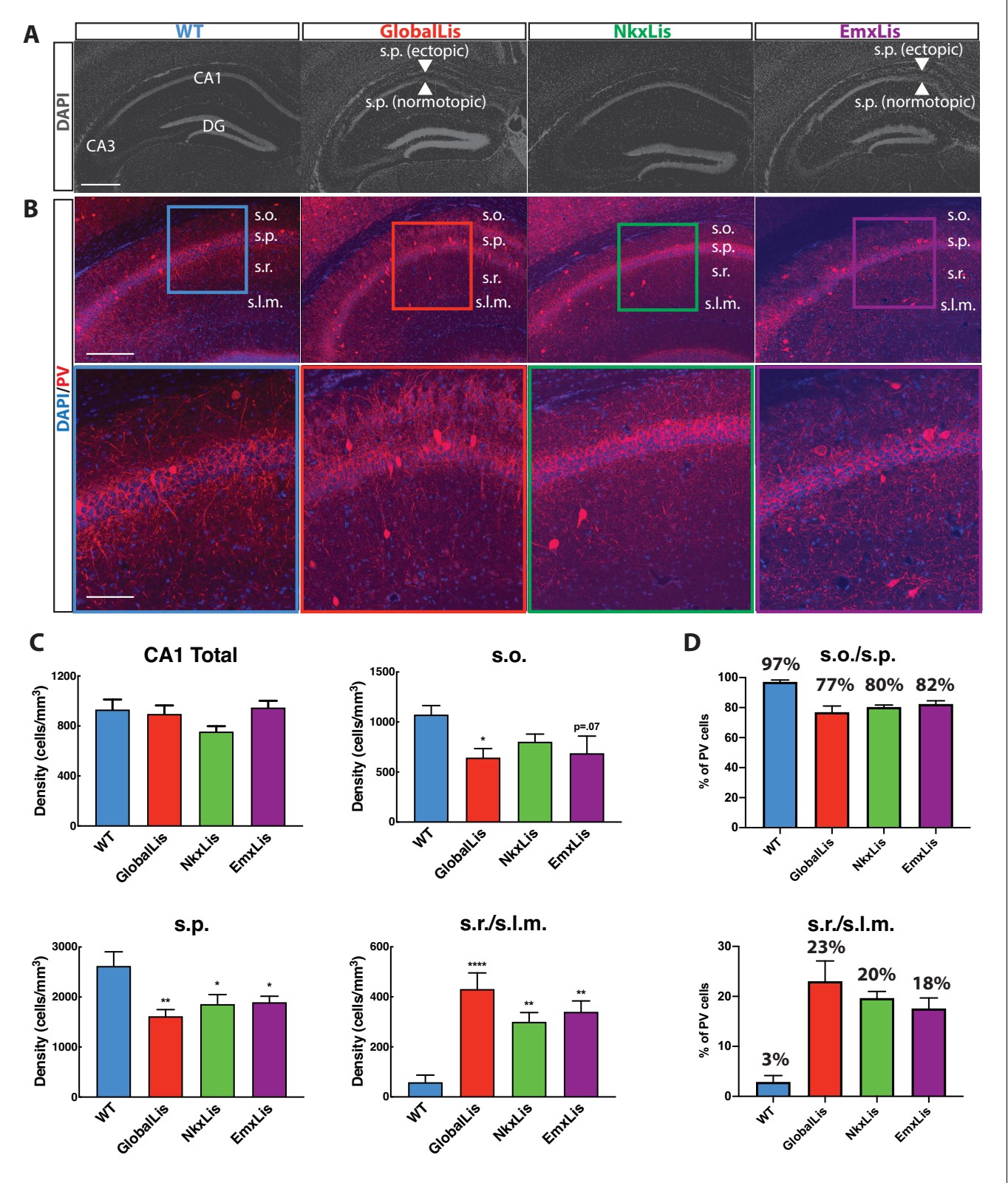

**Figure 1.** Cell-autonomous and non-autonomous effects of Pafah1b1 heterozygous loss on radial migration. (A) Images of DAPI-stained hippocampus of WT, GlobalLis (global mutation), NkxLis (IN-specific mutation), and EmxLis (PC-specific mutation) mice. (B) Images of PV- immunostaining in WT, GlobalLis, NkxLis, and EmxLis CA1. Bottom images show higher magnification of top images. (C) Quantified densities of PV+INTs in CA1 and each sublayer. Counting was performed on four hippocampal sections from each animal (n = 5 animals per genotype). For statistical analysis *p<0.05,

*Figure 1 continued on next page*

*Figure 1 continued*

**p<0.01, ***p<0.001, ****p<0.0001. (D) Percentage of PV+INTs in deep (s.o./s.p.) or superficial (s.r./s.l.m.) layers. Scale bar A = 600 µm, scale bar B (top) = 200 µm, scale bar B (bottom) = 120 µm.

The online version of this article includes the following source data for figure 1:

**Source data 1.** PV+INT density and % per layer.

To functionally characterize hippocampal PV+INTs we first examined the intrinsic electrophysiological properties of TdTomato+ (TdT) WT CA1 PV cells (*Figure 2A*). Accuracy of TdT labeling of PV+INTs was confirmed by quantifying the percentage of TdT and PV-immunostaining overlap: 92% of WT and 93% of GlobalLis hippocampal TdT+ cells were immunopositive for PV, and 97% of the immunopositive PV cells were labeled by TdT, enabling reliable targeting of PV+INTs (*Figure 2—figure supplement 1*).

WT fast-spiking PV+INTs have stereotypical intrinsic properties that include a low input resistance (75–90 MΩ), high rheobase (330–400 pA), high firing frequency at 2x (130–150 Hz) and 3x threshold (160–180 Hz), narrow action potential half-width (0.40–0.44 ms) and minimal spike-frequency-adaptation (0.75–0.85; *Table 1*). Surprisingly, we routinely observed an additional population (~15–20%) of WT TdT-labeled PV+INTs that did not possess stereotypical fast-spiking firing properties. This NFS subpopulation also displayed numerous intrinsic properties distinct from canonical FS cells, including a low firing frequency, high input resistance and low rheobase (i.e. *Figure 2A* top right). Clusters of atypical PV+INTs have been previously reported in subiculum ('quasi fast-spiking interneurons'; *Nassar et al., 2015*) and striatum ('fast-spiking-like cells'; *Muñoz-Manchado et al., 2018*).

To independently verify whether WT hippocampal PV+INTs could be functionally segregated into distinct clusters in an unbiased fashion, we performed principal component analysis (PCA) and K-means clustering using several key intrinsic physiological features (action potential half-width, firing frequency at 2x threshold, firing frequency at 3x threshold, adaptation ratio at 2x threshold, input resistance, rheobase, sag index). WT PV+INTs neatly parsed into two subtypes comprised of a large majority FS cohort, and a small minority atypical subset (*Figure 3A–D*). Due to their intrinsic physiological differences from FS cells and inability to sustain high frequencies of action potentials, for ease of discussion we designate this unique subpopulation of PV+INTs as 'non-fast-spiking' (NFS) cells. PV+NFS cells have lower firing frequencies at 2x (55–75 Hz) and 3x (85–105 Hz) threshold, lower adaptation ratios (0.60–0.75) broader action potential half-widths (0.55–0.65 ms), larger input resistances (120–170 MΩ), and lower rheobases (120–180 pA) than standard PV+FS cells (*Figure 3E*; *Table 1*).

With respect to their morphology, hippocampal PV+INTs are routinely parsed into three primary subtypes based on axonal arborization: basket cells (BCs; which target cell pyramidal cell (PC) bodies and proximal dendrites), axo-axonic cells (AACs; which target PC axon initial segments), and bistratified cells (BiCs; which target PC apical and basal dendrites; *Pelkey et al., 2017*). Post-hoc anatomical recoveries of recorded cells regularly revealed these three standard morphologies, and additionally a unique hippocampal PV+INT, which we designate 'radiatum-targeting cells' (RTC) as this subtype confines its axon to s.r. and is presumably a subtype of dendrite-targeting cell. *Figure 2A* shows typical morphologies, polar histograms of axonal and dendritic arbors and firing patterns at threshold (with expanded action potentials; red) and at 2x threshold current injection (phase plots in blue) of WT CA1 PV+INTs (from left to right: BC, AAC, BiC, PV+RTC).

Hippocampal layers differ in their composition of PV+INT morphophysiological subtypes. In WT deeper regions (s.o./s.p.) are populated primarily by PV+FS cells (~90%), consisting of all of the morphological forms (BC, AAC, BiC, RTC). In contrast, all PV+NFS cells residing in s.o./s.p. had BC morphologies (*Figure 3F–G*). The small number of PV+FS and NFS cells found in superficial layers (s.r.) all had RTC morphology (*Figure 3F–G*). In summary, WT hippocampal CA1 PV+INTs consist of two physiological (FS and NFS) and four morphological subtypes (BC, AAC, BiC, RTC), and the overwhelming majority of PV+INTs are found in deep hippocampal layers, consistent with previous reports (*Pelkey et al., 2017*).

Using the same strategy, we next targeted PV+INTs in the GlobalLis mouse line and utilized PCA to cluster GlobalLis PV+INTs by their intrinsic physiological properties. Unlike WT PV+INTs, GlobalLis cells segregated into three clusters. (*Figure 4A–D*). In addition to the FS and NFS cell clusters, a

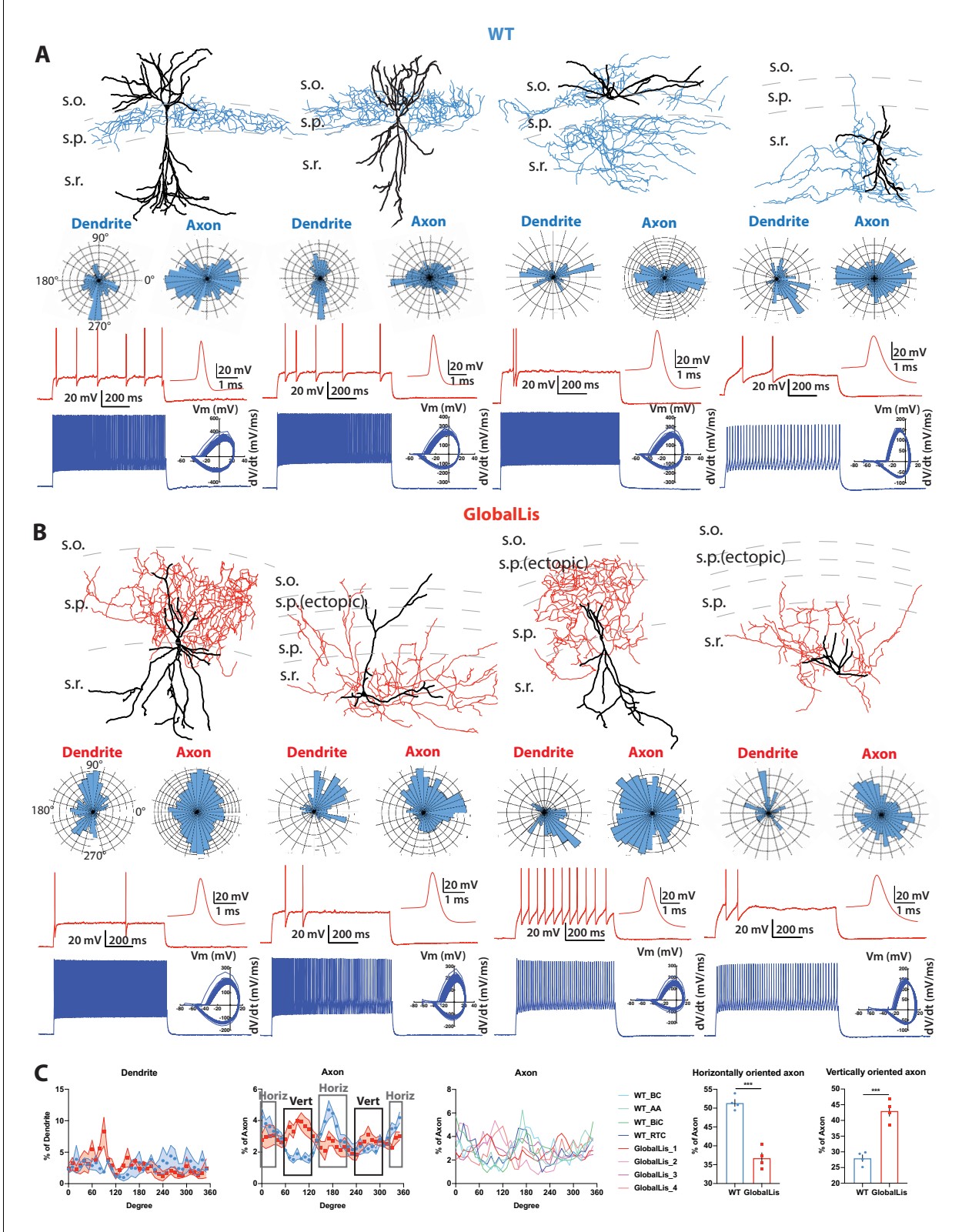

**Figure 2.** Morphological reconstructions and firing profiles of WT and GlobalLis PV+INTs. (**A**) Examples of WT PV+INTs with reconstructed cells on top (dendrite in black, axon in blue), polar histograms of dendrite and axonal arbor orientation in the middle and firing traces on bottom. Firing in response to threshold current is shown in red, with the first action potential shown expanded on the right. The blue trace displays firing at 2x threshold current and shows a phase plot on the right. Cell types from left to right: basket cell, axo-axonic cell, bistratified cell, radiatum-targeting cell. (**B**) Examples of

*Figure 2 continued on next page*

*Figure 2 continued*

GlobalLis PV+INTs. Dendrite is shown in black and axon in red. Cells can no longer be neatly parsed into the four morphological PV+INT subtypes found in WT CA1. (C) Left, polarity/degree preferences for WT and GlobalLis PV+INT dendrites and axons. Right, analysis of WT and GlobalLis axonal horizontal or vertical orientation preferences.

The online version of this article includes the following source data and figure supplement(s) for figure 2:

**Source data 1.** Axon and dendrite polar histograms.
**Figure supplement 1.** PV-TdTomato/PV-IHC colocalization.
**Figure supplement 1—source data 1.** PV-TdTomato/PV-IHC co-localization.

**Table 1.** Membrane, firing and morphological properties of PV+INTs.

| | WT FS 67 cells | GlobalLis FS 20 cells | NkxLis FS 12 cells | EmxLis FS 30 cells | WT IS 0 cells | GlobalLis IS 21 cells | NkxLis IS 21 cells | EmxLis IS 0 cells | WT NFS 16 cells | GlobalLis NFS 4 cells | NkxLis NFS 8 cells | EmxLis NFS 8 cells |
|---|---|---|---|---|---|---|---|---|---|---|---|---|
| Input resistance (MΩ) | 82.5 ± 3.6 | 63.5 ± 4.2 | 61.1 ± 6.3 | 75.0 ± 4.7 | N/a | 88.5 ± 4.9 | 85.5 ± 7.0 | N/a | 143.1 ± 11.9 | 132.0 ± 14.4 | 200.7 ± 17.0 | 173.2 ± 9.0 |
| Rheobase (pA) | 364.2 ± 17.3 | 457.5 ± 22.2 | 533.3 ± 42.3 | 463.3 ± 24.8 | n/a | 340.5 ± 20.0 | 342.9 ± 21.1 | n/a | 153.1 ± 14.8 | 162.5 ± 20.0 | 125.0 ± 13.4 | 143.8 ± 11.3 |
| Firing Freq 2x Threshold (Hz) | 137.6 ± 4.7 | 168.5 ± 11.2 | 173.2 ± 14.8 | 155.4 ± 7.4 | n/a | 106.2 ± 4.8 | 95.6 ± 3.5 | n/a | 66.0 ± 4.5 | 57.0 ± 11.6 | 72.3 ± 4.4 | 58.0 ± 3.5 |
| Adaptation Ratio 2x Threshold | 0.79 ± 0.02 | 0.80 ± 0.03 | 0.72 ± 0.04 | 0.71 ± 0.02 | n/a | 0.71 ± 0.03 | 0.65 ± 0.03 | n/a | 0.67 ± 0.03 | 0.79 ± 0.05 | 0.79 ± 0.06 | 0.52 ± 0.05 |
| Firing Freq 3x Threshold (Hz) | 170.9 ± 5.1 | 216.5 ± 11.5 | 213.0 ± 15.3 | 195.7 ± 10.3 | n/a | 130.2 ± 6.0 | 131.1 ± 5.1 | n/a | 96.1 ± 4.6 | 74.5 ± 15.1 | 100.8 ± 5.7 | 77.8 ± 6.9 |
| Adaptation Ratio 3x Threshold | 0.78 ± 0.01 | 0.82 ± 0.03 | 0.71 ± 0.05 | 0.71 ± 0.02 | n/a | 0.67 ± 0.02 | 0.57 ± 0.03 | n/a | 0.65 ± 0.03 | 0.64 ± 0.10 | 0.73 ± 0.05 | 0.47 ± 0.05 |
| AP Threshold (mV) | −39.9 ± 0.7 | −41.2 ± 1.2 | −39.9 ± 1.3 | −37.8 ± 1.1 | n/a | −40.4 ± 1.1 | −35.9 ± 1.2 | n/a | −40.6 ± 1.2 | −38.6 ± 2.8 | −36.8 ± 1.2 | −36.8 ± 1.3 |
| AP Half-width (ms) | 0.42 ± 0.01 | 0.34 ± 0.01 | 0.35 ± 0.01 | 0.36 ± 0.01 | n/a | 0.48 ± 0.01 | 0.44 ± 0.01 | n/a | 0.59 ± 0.02 | 0.64 ± 0.07 | 0.56 ± 0.04 | 0.58 ± 0.04 |
| AP Amplitude (mV) | 61.1 ± 1.0 | 61.2 ± 2.7 | 55.8 ± 3.0 | 57.5 ± 2.0 | n/a | 62.1 ± 2.3 | 53.6 ± 2.0 | n/a | 65.4 ± 3.2 | 64.8 ± 7.0 | 55.1 ± 5.7 | 64.9 ± 4.8 |
| AP Max Rise Slope (mV/ms) | 259.9 ± 5.2 | 248.3 ± 13.5 | 268.6 ± 11.7 | 217.0 ± 12.1 | n/a | 261.4 ± 10.7 | 231.0 ± 10.3 | n/a | 228.6 ± 11.5 | 210.3 ± 25.0 | 197.9 ± 18.3 | 241.0 ± 26.0 |
| AP Max Decay Slope (mV/ms) | −188.7 ± 5.5 | −176.0 ± 14.3 | −201.6 ± 9.1 | −211.4 ± 12.6 | n/a | −198.1 ± 12.6 | −149.2 ± 7.1 | n/a | −136.5 ± 11.5 | −131.3 ± 20.6 | −119.2 ± 14.7 | −125.4 ± 16.8 |
| AHP Amplitude (mV) | −16.8 ± 0.6 | −17.5 ± 0.9 | −15.3 ± 0.9 | −16.7 ± 0.7 | n/a | −16.3 ± 0.8 | −16.0 ± 1.0 | n/a | −15.7 ± 1.2 | −16.9 ± 1.1 | −17.0 ± 1.2 | −11.3 ± 1.7 |
| Membrane Time Constant (ms) | 8.5 ± 0.5 | 7.5 ± 0.4 | 6.7 ± 0.4 | 8.6 ± 0.4 | n/a | 8.7 ± 0.4 | 8.5 ± 0.7 | n/a | 10.9 ± 0.6 | 12.9 ± 0.6 | 14.8 ± 1.6 | 17.1 ± 1.0 |
| Membrane Capacitance (pF) | 104.4 ± 5.0 | 117.7 ± 6.6 | 116.8 ± 18.6 | 123.9 ± 7.7 | n/a | 103.7 ± 8.6 | 105.7 ± 8.3 | n/a | 91.3 ± 6.4 | 88.6 ± 6.4 | 73.8 ± 4.8 | 97.0 ± 4.3 |
| Sag Index | 0.90 ± 0.01 | 0.91 ± 0.01 | 0.95 ± 0.01 | 0.92 ± 0.01 | n/a | 0.90 ± 0.01 | 0.91 ± 0.01 | n/a | 0.8 ± 0.03 | 0.82 ± 0.04 | 0.86 ± 0.02 | 0.85 ± 0.02 |
| Total Sholl Intersections (Dendrite) | 48 ± 7 (19 cells) | 65 ± 6 (6 cells) | 43 ± 8 (8 cells) | 56 ± 7 (11 cells) | n/a | 43 ± 7 (7 cells) | 61 ± 11 (8 cells) | n/a | 26 ± 4 (10 cells) | 28 ± 8 (2 cells) | 34 ± 11 (3 cells) | 38 ± 10 (4 cells) |
| Total Sholl Intersections (Axon) | 222 ± 22 | 271 ± 44 | 206 ± 44 | 324 ± 51 | n/a | 224 ± 56 | 224 ± 41 | n/a | 131 ± 15 | 133 ± 3 | 97 ± 21 | 95 ± 26 |
| Postnatal age at time of recording (days) | 29 ± 1 | 26 ± 1 | 29 ± 1 | 29 ± 2 | n/a | 26 ± 1 | 27 ± 1 | n/a | 28 ± 1 | 25 ± 3 | 27 ± 1 | 28 ± 3 |

The online version of this article includes the following source data for Table 1:
**Source data 1.** Morphophysiological properties of PV+INTs.

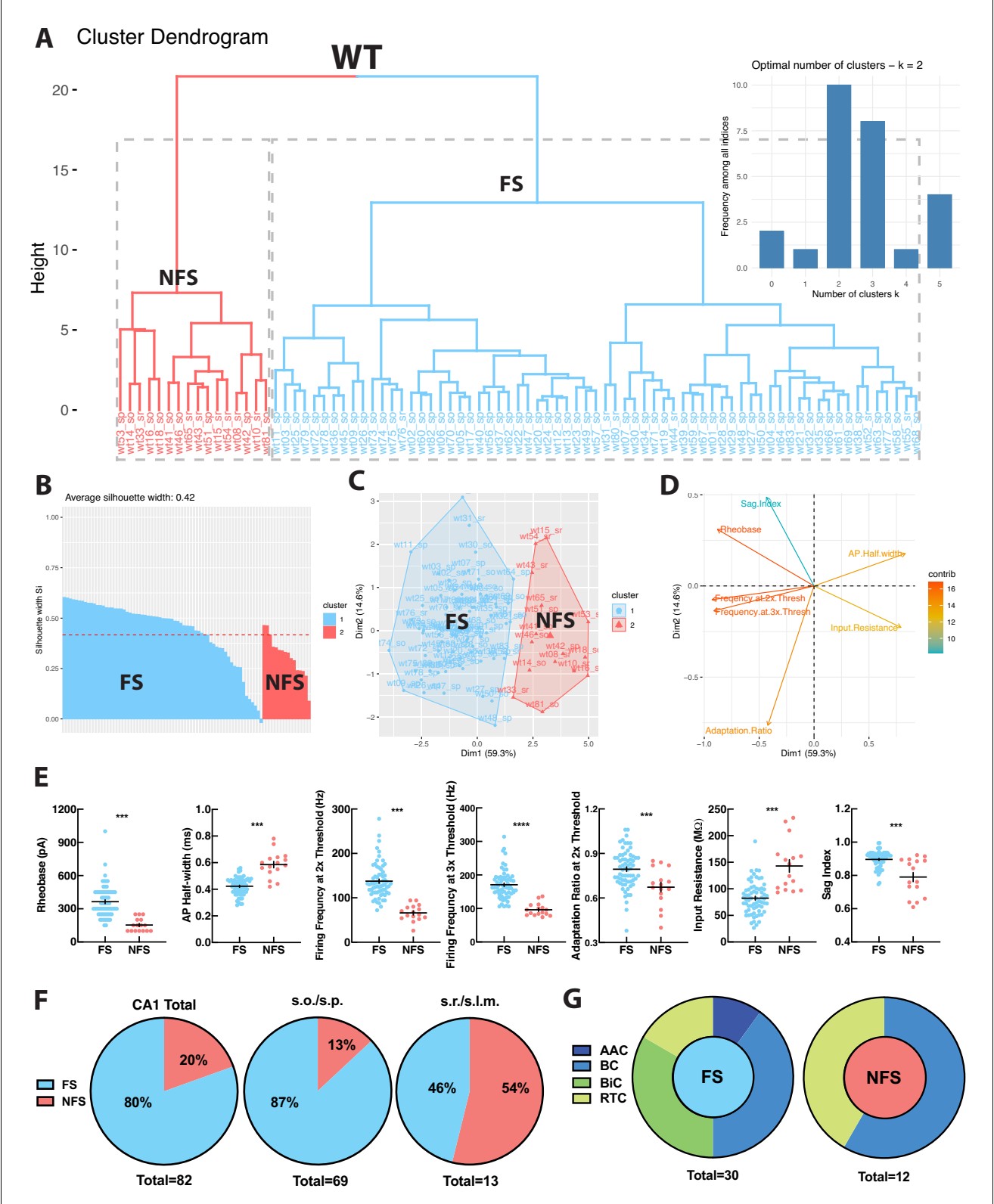

**Figure 3.** WT PV+INTs consist of two physiological subtypes: FS and NFS cells. (**A**) Unbiased cluster analysis dendrogram displays 83 PV+INTs sorted two clusters that represent fast-spiking (FS) and non-fast-spiking (NFS) cells. Inset indicates the optimal number of clusters, determined using NbClust function in R is 2 (FS and NFS). (**B**) Silhouette plot of the FS/NFS clusters. (**C**) Cluster plot of the same 83 PV+INTs. Nonoverlap indicates clear segregation of FS/NFS clusters. (**D**) Contributions of each intrinsic physiological property used in the cluster analysis. (**E**) Plots displaying and FS/NFS

*Figure 3 continued on next page*

*Figure 3 continued*

cell intrinsic properties. For statistical analysis *p<0.05, **p<0.01, ***p<0.001, ****p<0.0001. (F) Percentages of FS/NFS cells in CA1 and in each sublayer. (G) Distributions of FS and NFS cell morphological subtypes. FS cells consisted of BCs, AACs, BiCs, and RTCs, whereas all recovered NFS cells were identified as BCs or RTCs.

The online version of this article includes the following figure supplement(s) for figure 3:

**Figure supplement 1.** Cluster analysis and morphophysiology of EmxLis PV+INTs.

third entirely new cluster of PV+INTs emerged (*Figure 4A–D*). GlobalLis FS cells had identical intrinsic properties as WT FS cells with the exception of shorter action potential half-widths (0.32–037 ms vs 0.41–0.44 ms; *Table 1*). GlobalLis NFS cells were indistinguishable from WT NFS cells. A number of intrinsic physiological properties of the third group of cells fell between those of FS and NFS cells, including firing frequency at 2x (90–110 Hz) and 3x threshold (120–140 Hz), adaptation ratio (0.65–0.85), AP half-width (0.45–0.50 ms), and input resistance (80–100 MΩ; *Figure 4E*; *Table 1*). Consequently, we refer to this emergent physiological PV+INT subtype as 'intermediate spiking' (IS) cells. Of particular interest 47% of all GlobalLis PV+INTs consisted of the IS subtype and only 44% consisted of FS cells (*Figure 4F*) compared the 80% observed in WT CA1 PV+INTs. Within CA1, superficial layers had proportionally more IS cells (~60%) than deeper layers (~25%; *Figure 4F*). A Fisher exact test confirms differences in WT and GlobalLis composition of subtypes of CA1 PV+INTs (p<0.0001; *Figure 4G*).

In addition to somatic misplacement and physiological disruption, the morphological development of GlobalLis PV+INTs is radically disrupted. Many cells did not resemble stereotypical morphologies of any WT PV+INT subtype (*Figure 2A*) and often had ectopic axonal branching that extended in all directions, rather than forming the tight plexus seen in WT. Other mutant PV+INTs took on combinations of hippocampal PV cell features including bistratified-like cells with baskets, and radiatum-targeting cells with axons that also extended into the s.l.m. (*Figure 2B*). Analysis of PV+INT polar histograms indicates that GlobalLis cells have alterations in the direction of axonal (but not dendritic) growth. GlobalLis PV+INTs project more axon in vertical (i.e. from s.o. to s.l.m.) than horizontal (i.e. from CA3 to subiculum) directions, a stark contrast to WT cells, which have more horizontally oriented axon (*Figure 2C*). In conclusion, global mutations to *Pafah1b1* disrupt morphophysiological identity in a large number of PV+INTs; however, overall axonal and dendritic growth is not inhibited.

## Cell-autonomous *Pafah1b1* mutation within interneurons disrupts PV +INT development

We next recorded from PV+INTs in both the EmxLis and NkxLis mutants to determine if the disruption of PV+INT morphophysiological development emerges from interactions in a malformed hippocampus (EmxLis, *Figure 1B–D*) or from selective disruption of inhibitory interneuron intrinsic developmental programs (NkxLis, *Figure 1B–D*). PCA of intrinsic physiological properties of PV +INTs from the EmxLis hippocampus identified two clusters (corresponding to FS and NFS), consistent with WT (*Figure 3—figure supplement 1A*). To increase the power of the PCA and confirm accurate clustering, we combined and analyzed the WT and EmxLis datasets together. Combining WT and EmxLis cells resulted in identical clusters of FS and NFS cells, indicating the ease of identifying normally developed FS and NFS cell types (*Figure 3—figure supplement 1B*). The total ratio of FS/NFS cells in CA1 as well as in each individual layer remains relatively unchanged in the EmxLis hippocampus (*Figure 3—figure supplement 1C*). Furthermore, morphological development of PV +INTs was relatively unaffected by non-autonomous *Pafah1b1* mutation: PV+ cells developed into identifiable BCs, BiCs, AACs, and RTCs (*Figure 3—figure supplement 1D*). In conclusion, in EmxLis mutants PV+INTs develop into canonical morphophysiological subtypes despite disorganized pyramidal cell layers.

We next examined PV+INTs in the NkxLis (interneuron-specific mutation) hippocampus and used PCA to analyze the physiological properties. Similar to the GlobalLis mutant, we detected 3 clusters of PV+INTs, however clustering accuracy was not optimal (i.e. occasional IS cells were classified as NFS cells etc; *Figure 4—figure supplement 1A*). To increase analytical power and improve clustering accuracy, we combined the NkxLis and GlobalLis datasets and again found three clusters: FS, IS,

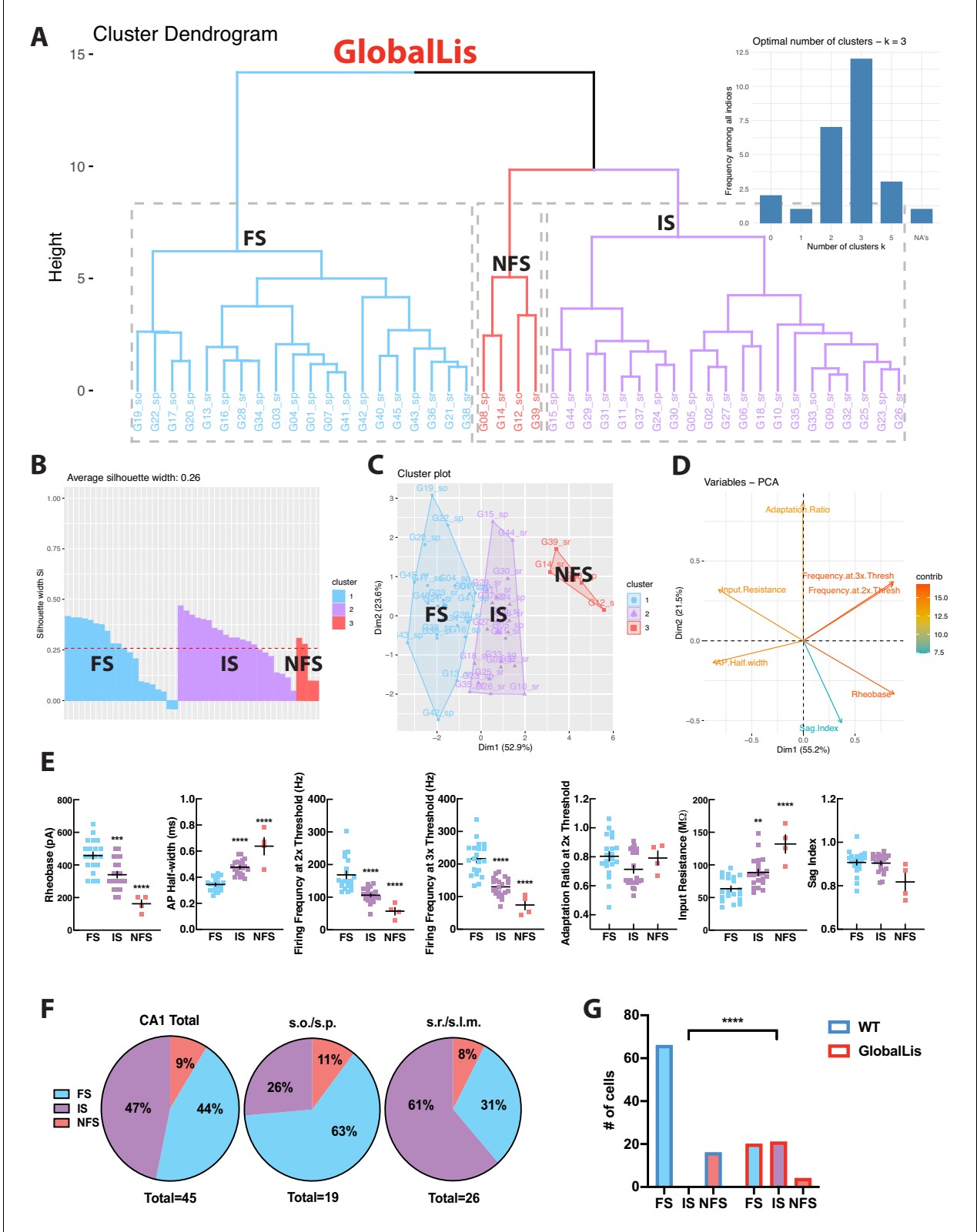

**Figure 4.** GlobalLis PV+INTs consist of three physiological subtypes: FS, IS, and NFS cells. (**A**) Unbiased cluster analysis indicates emergence of an additional cluster in GlobalLis CA1, displayed in the dendrogram of 45 PV+INTs. Inset indicates optimal detection of three clusters (FS, IS, NFS). (**B**) Silhouette plot of the FS/IS/NFS clusters. (**C**) Cluster plot of the same 45 PV+INTs. (**D**) Contributions of each intrinsic physiological property used in the cluster analysis. (**E**) Plots displaying FS/IS/NFS cell intrinsic properties. For statistical analysis *p<0.05, **p<0.01, ***p<0.001, ****p<0.0001. (**F**)

*Figure 4 continued on next page*

*Figure 4 continued*

Percentage of FS/IS/NFS cells in CA1 and in each sublayer. (**G**) Fisher exact test confirms differences in WT and GlobalLis composition of CA1 PV+INT subtypes.

The online version of this article includes the following source data and figure supplement(s) for figure 4:

**Figure supplement 1.** Cluster analysis and morphophysiology of NkxLis PV+INTs.
**Figure supplement 2.** Comparison of K-means and Ward clustering.
**Figure supplement 3.** Sholl analysis of PV+INT physiological subtypes.
**Figure supplement 3—source data 1.** Sholl analysis of PV+INTs.
**Figure supplement 4.** Membrane and firing properties of PV+INT physiological subtypes.

and NFS cells (*Figure 4—figure supplement 1B*). The novel IS cell subtype accounts for ~50% of total PV+INTs in both the NkxLis and GlobalLis genotypes, with more IS cells found in superficial hippocampal layers (*Figure 4—figure supplement 1C*). Thus, with cell-autonomous *Pafah1b1* mutations within interneurons, the overall distribution of PV+INT physiological subtypes shifts away from the 80% FS, 20% NFS distribution in WT (and EmxLis) to 44% FS, 56% IS/NFS in the GlobalLis mutant and 29% FS, 71% IS/NFS in the NkxLis mutant. Despite the normal lamination of pyramidal cell layers in the NkxLis hippocampus, the morphology of NkxLis PV+INTs appear similar to the GlobalLis mutants; cells take on hybrid combinations of morphological features and distinguishing meaningful subtypes becomes nontrivial, if not impossible (*Figure 4—figure supplement 1D*).

To validate the accuracy of our cluster analysis, we performed additional clustering using Ward's Method and compared the cluster identity of cells from Ward's and K-means clustering algorithms. In the WT and EmxLis genotypes, Ward clustering yielded identical results as K-means clustering. In the GlobalLis and NkxLis genotypes, Ward clustering was very similar to the K-means clustering, together confirming the sorting accuracy (*Figure 4—figure supplement 2*).

Sholl analyses of digitally reconstructed axonal and dendritic arbors or recorded PV+INTs revealed that FS cells in all genotypes and GlobalLis and NkxLis IS cells had larger and more complex axonal and dendritic trees than NFS cells of all genotypes (*Figure 4—figure supplement 3*). The smaller axonal arbors of NFS cells (WT: 131 ± 15 Sholl intersections, GlobalLis: 133 ± 3 intersections, NkxLis: 97 ± 21 intersections, EmxLis: 95 ± 26 intersections) relative to FS (WT: 222 ± 22 Sholl intersections, GlobalLis: 271 ± 44 intersections, NkxLis: 206 ± 44 intersections, EmxLis: 324 ± 51 intersections) and IS cells (GlobalLis: 224 ± 56 intersections, NkxLis: 224 ± 41 intersections) may imply lower synaptic connectivity of this subtype (*Figure 4—figure supplement 3C*; *Table 1*).

Finally, we tested whether subtypes of PV+INTs (FS, IS, NFS) shared intrinsic physiological properties across genotypes. Similar to our observation of shorter action potential half-widths in GlobalLis FS cells, half-width was also shorter than WT in NkxLis (0.32–0.38 ms) and EmxLis (0.34–0.39 ms) genotypes (*Table 1*). All other intrinsic properties of PV+INT physiological subtypes were preserved across all genotypes (*Figure 4—figure supplement 4*; *Table 1*). Taken together, these results indicate that while both cell-autonomous and non-autonomous *Pafah1b1* mutations can disrupt somatic positioning, only cell-autonomous mutations within interneurons perturb the morphophysiological identity of PV+INTs.

## PV+INT microcircuit rearrangements in the GlobalLis hippocampus

We next investigated the impact of *Pafah1b1* haploinsufficiency on microcircuit formation by examining excitatory input and inhibitory output of PV+INTs. We used whole-cell recordings to record spontaneous excitatory post synaptic currents (sEPSCs) on WT and GlobalLis PV+INTs and found several differences in the frequency and kinetics of excitatory currents. WT and GlobalLis FS cells had higher sEPSC frequencies (26 ± 4 Hz and 27 ± 6 Hz, respectively) and shorter EPSC decay time constants (1.6 ± 0.2 ms, 1.1 ± 0.1 ms) than NFS (WT: 7 ± 6 Hz, 4.1 ± 0.8 ms; GlobalLis: 4 ± 2 Hz, 3 ± 1.5 ms) and IS cells (5 ± 1 Hz, 1.7 ± 0.3 ms; *Figure 5—figure supplement 1*). These data suggest that FS cells in WT and the GlobalLis mouse remain the primary PV_ interneuron targets for excitatory input in the CA1 hippocampus.

We next assayed PV+INT inhibitory output by using dual whole-cell recordings between synaptically coupled pairs of PV+INTs and CA1 pyramidal cells (*Figure 5A–B*). In general, the connection probability of WT FS cells onto PCs was higher than that observed between NFS cells and PCs (32%

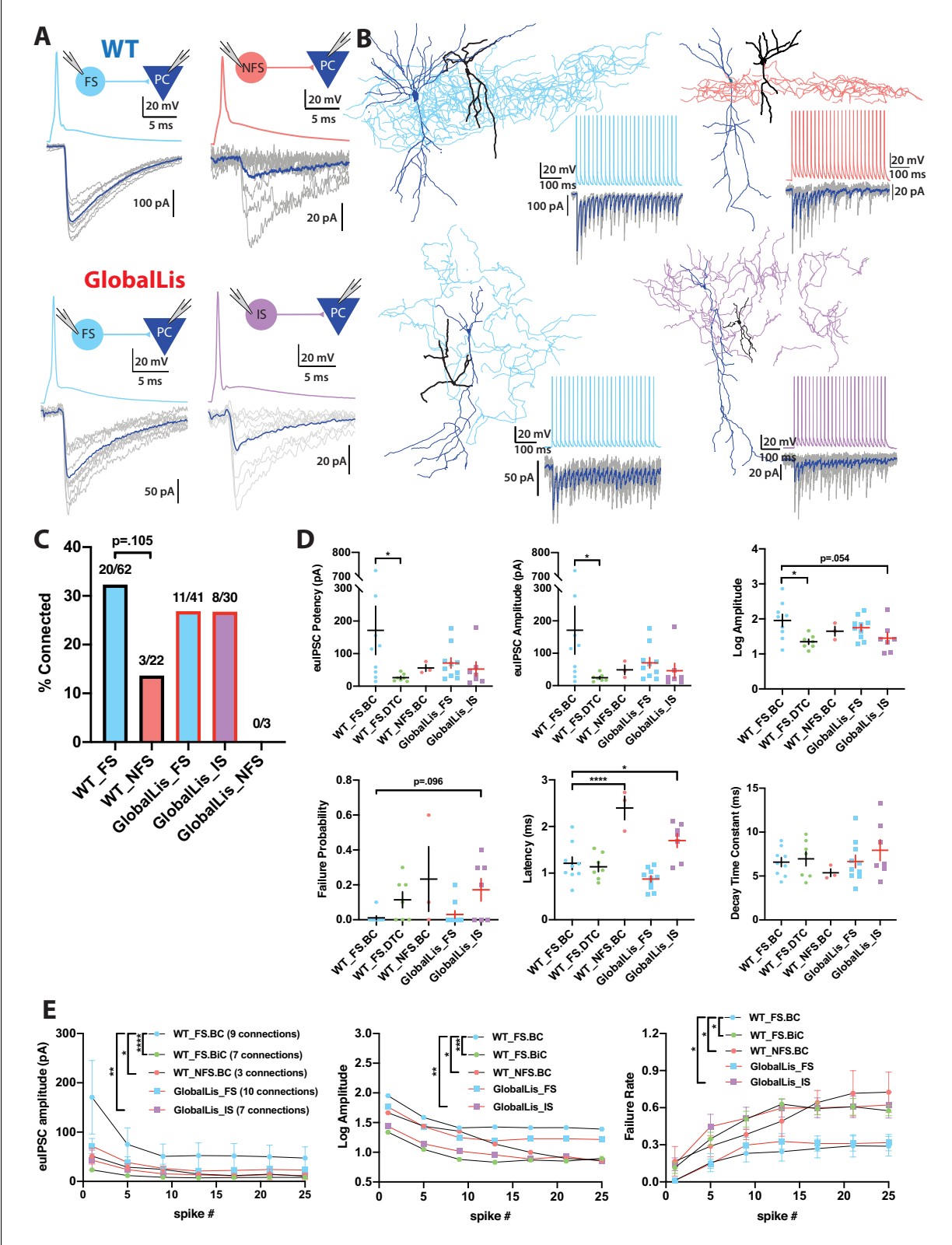

**Figure 5.** Connectivity and microcircuitry of PV+INTs. (A) (i) Examples of paired whole-cell recordings between synaptically connected presynaptic PV +INTs (top traces) and postsynaptic CA1 PCs (bottom traces). 10 individual traces (grey) and an averaged trace (navy) are show for the postsynaptic cells. Note that a high [Cl-] internal solution was used in the PCs, resulting in GABAergic inward currents when PCs were clamped at −70 mV. (B) Reconstructions of PV+INTs (dendrite/cell body in black, axon in light blue, pink or purple) and PCs (dendrite/cell body in navy) and sample traces of a
*Figure 5 continued on next page*

*Figure 5 continued*

50 Hz, 25 pulse stimulation. (**C**) Connectivity of PV+INT to PCs. Note the lower connectivity of NFS cells. (**D**) Unitary transmission properties between PV+INTs and PCs. (**E**) Short-term plasticity of PV+INT microcircuits during a 50 Hz, 25 pulse train. For statistical analysis *p<0.05, **p<0.01, ***p<0.001, ****p<0.0001.

The online version of this article includes the following source data and figure supplement(s) for figure 5:

**Source data 1.** PV+INT unitary transmission properties.
**Figure supplement 1.** Analysis of excitatory input to PV+INTs.
**Figure supplement 1—source data 1.** sEPSC properties of PV+INTs.

vs 14% connected; *Figure 5C*). Connectivity rates of GlobalLis FS cells (27%; 11/41 cells) were similar to WT FS cells. IS cells had a 27% connection probability (8/30 cells) onto PCs and 0/3 NFS cells were connected to PCs (*Figure 5C*).

WT FSBCs connected to PCs with high potency synapses (170 ± 75 pA) which had an extremely low neurotransmission failure probability (0.01 ± 0.01) and quick latency to release (1.2 ± 0.1 ms). Compared to FSBCs, WT FSBiC-PC connections had weaker synapses (25 ± 5 pA), slightly higher failure probability (0.11 ± 0.05) and a nearly identical latency (1.1 ± 0.1 ms), while NFSBCs also had weaker connections (50 ± 15 pA), they also had higher failure probability (0.23 ± 0.19) but longer transmission latency (2.4 ± 0.3 ms). In the GlobalLis mutant, FS-PC connections were weaker (70 ± 15 pA) than WT FSBCs, but not significantly different when we pooled the datasets of WT FSBC and FSBiC-PC connections (110 ± 45 pA). GlobalLis FS cell failure probability (0.03 ± 0.02) and latency (0.9 ± 0.1 ms) were indistinguishable from WT FS cells. Similar to NFS cells, IS cells had higher failure probability (0.17 ± 0.06) and longer latency (1.7 ± 0.2 ms), contrasting FS-PC connections (*Figure 5D*; *Table 2*). Unfortunately, no GlobalLis NFS-PC connections were obtained.

We next investigated short-term transmission dynamics using high-frequency trains of presynaptic action potentials (*Figure 5B*). All PV-PC pairs showed characteristic synchronous neurotransmitter release and marked synaptic depression across a 50 Hz, 25 pulse train. In WT FSBCs, by the end of the train, uIPSC amplitude at the end of the 50 Hz train was reduced by 70% and the failure rate increased to 29%. Transmission at WT FSBiC connections fell by 61% and possessed a higher failure rate (58%). Unitary connections at WT NFS cells fell to 82% of the first pulse and the failure rate was 73%. In the GlobalLis mutant, FS-PC connection strength dropped by 70% and the failure rate increased to 32% by the end of the train; values close to those seen in WT FS cells. Similarly, Global-Lis IS connections dropped by 71% but the failure rate increased to 62%, similar to NFS cells (*Figure 5E*). Collectively, these results confirm that at the monosynaptic level, both the unitary synaptic amplitudes and the short-term dynamics of transmission are unchanged in GlobalLis FS-PC connections; however, IS-PC synapses resemble those of NFS cells.

**Table 2.** Unitary transmission properties by PV+INT subtype.

|  | WT FS.BC 9 connections | WT FS.DTC 7 connections | WT NFS.BC 3 connections | GlobalLis FS connections | GlobalLis IS connections |
|---|---|---|---|---|---|
| euIPSC potency (pA) | 171 ± 74 | 26 ± 5 | 56 ± 10 | 71 ± 16 | 56 ± 20 |
| euIPSC Amplitude (pA) | 171 ± 74 | 24 ± 4 | 49 ± 15 | 66 ± 15 | 48 ± 20 |
| Failure probability | 0.01 ± 0.01 | 0.11 ± 0.05 | 0.23 ± 0.19 | 0.03 ± 0.02 | 0.17 ± 0.06 |
| Latency (ms) | 1.2 ± 0.1 | 1.1 ± 0.1 | 2.4 ± 0.3 | 0.9 ± 0.1 | 1.6 ± 0.1 |
| Decay time constant (ms) | 6.6 ± 0.5 | 7.0 ± 0.8 | 5.4 ± 0.4 | 6.5 ± 0.7 | 7.7 ± 1.1 |
| Postnatal age at time of recording (days) | 35 ± 3 | 30 ± 4 | 27 ± 1 | 30 ± 1 | 27 ± 2 |

The online version of this article includes the following source data for Table 2:
**Source data 1.** PV+INT euIPSC properties.

## Depolarization block of action potential firing is a common feature of IS/NFS but not FS cells

PV+INTs are critical regulators of network excitability, such that their rapid action potential and transmission kinetics act to generate both feedforward and feedback inhibition and local oscillations, as well as preventing cortical network activity from disintegrating into electrographic events that are the underpinnings of seizure activity. Recent evidence suggests that in epileptic human patients, PV +INT function becomes compromised, and overactivation of PV+INTs can drive action potentials into depolarization block, causing a temporary loss of their action potential initiation, consequently resulting in a net decrease in inhibitory tone, and precipitation of seizure propagation (*Sudhakar et al., 2019*; *Ahmed et al., 2020*).

We were struck by our observation that the majority of PV+INTs in the GlobalLis mouse are IS/NFS cells (56%) compared to FS, which represent only 44% of the total PV+INT population (cf. 80% in WT). Both IS and NFS cells possess lower maximal firing frequencies and longer duration action potentials compared to FS cells. This relative shift in the overall PV+ cell population and their network dynamics may compromise PV+INT network control over network excitability in the *Pafah1b1* mutant and promote lower seizure thresholds observed in previous studies (*Fleck et al., 2000*).

FS PV+INTs are endowed with both voltage-gated $Na^+$ and $K^+$ channels that enable rapid action potential repolarization and conductance deinactivation, permitting repetitive high-frequency firing with little firing accommodation and a resistance to depolarizing block (*Rudy and McBain, 2001*; and *Hu et al., 2014*; *Pelkey et al., 2017*). However, it is unclear whether either NFS or IS PV+INTs share the same resistance to firing accommodation or depolarization block of AP firing during sustained excitation.

Depolarization block susceptibility of WT and GlobalLis PV+INTs was first assayed by injecting PV +INTs with increasing suprathreshold currents (500 ms duration; *Figure 6A*). Due to their low input resistance, FS cells in both WT and GlobalLis can sustain large current injections before action potentials enter into a depolarization block, with the vast majority of cells (70–77%) fully retaining the ability to fire at high frequencies (>250 Hz) even after a maximal current injection of 1500 pA (*Figure 6B*). In contrast, 100% of NFS in both WT and GlobalLis mice and 92% of GlobalLis IS cells were driven into depolarizing block, and required less current to do so (respectively 690 ± 110 pA, 700 ± 200 pA, 990 ± 80 pA) than the 23–30% of FS cells that we did observe to enter depolarizing block (WT: 1230 ± 180 pA, GlobalLis: 1280 ± 110 pA; *Figure 6B*).

In our previous study (*Fleck et al., 2000*), we demonstrated that the CA1 hippocampus in *Pafah1b1* mutant mice had a lowered threshold for electrographic events in the 8.5 mM K+ mouse model of epilepsy (*Traynelis and Dingledine, 1988*). In this model, a modest 5 mM elevation of extracellular K+ promotes cellular depolarization coupled to a change in the $E_K$ of +23 mV. Previously, we did not explore the cellular underpinnings for this change in seizure threshold but now consider that the change in extracellular K+ may drive the expanded non-canonical PV+INT network into a more depolarized state that reduces their propensity for sustained action potential activity. Therefore, we revisited this mouse model of epilepsy to determine whether either hippocampal pyramidal cells or PV+INTs enter depolarization block under conditions of elevated extracellular K known to drive electrographic activity.

We recorded evoked and spontaneous action potentials in WT and GlobalLis PV+INTs and pyramidal neurons during a 15-min exposure period to elevated extracellular [K+] (8.5 mM; *Figure 6C*). We found that no recorded WT FS cells (0/10) and only 1/7 GlobalLis FS cells (14%) lost the ability to sustain repetitive action potential firing in the face of elevated [K+]. In contrast, 25% of IS cells (2/8), and 33% of both WT (1/3) and GlobalLis NFS cells (1/3) entered depolarization block following exposure to elevated [K+]. No WT (0/5) or GlobalLis (0/4) hippocampal pyramidal neurons lost the capability to repetitively fire action potentials in 8.5 mM [K+] (*Figure 6D*). Taken together, these data demonstrate that relative to FS cells, IS and NFS PV+INTs have a strong propensity to enter depolarization block in response to elevated [K+] and suprathreshold depolarizing current injection.

## Single-cell nuclear RNAsequencing (snRNA-seq) reveals molecular changes in response to *Pafah1b1* loss

Finally, we used snRNA-seq sequencing to understand how *Pafah1b1* haploinsufficiency impacts gene expression in the subtypes of PV+INTs. GlobalLis mice were crossed to PVCre;Sun1-GFP mice

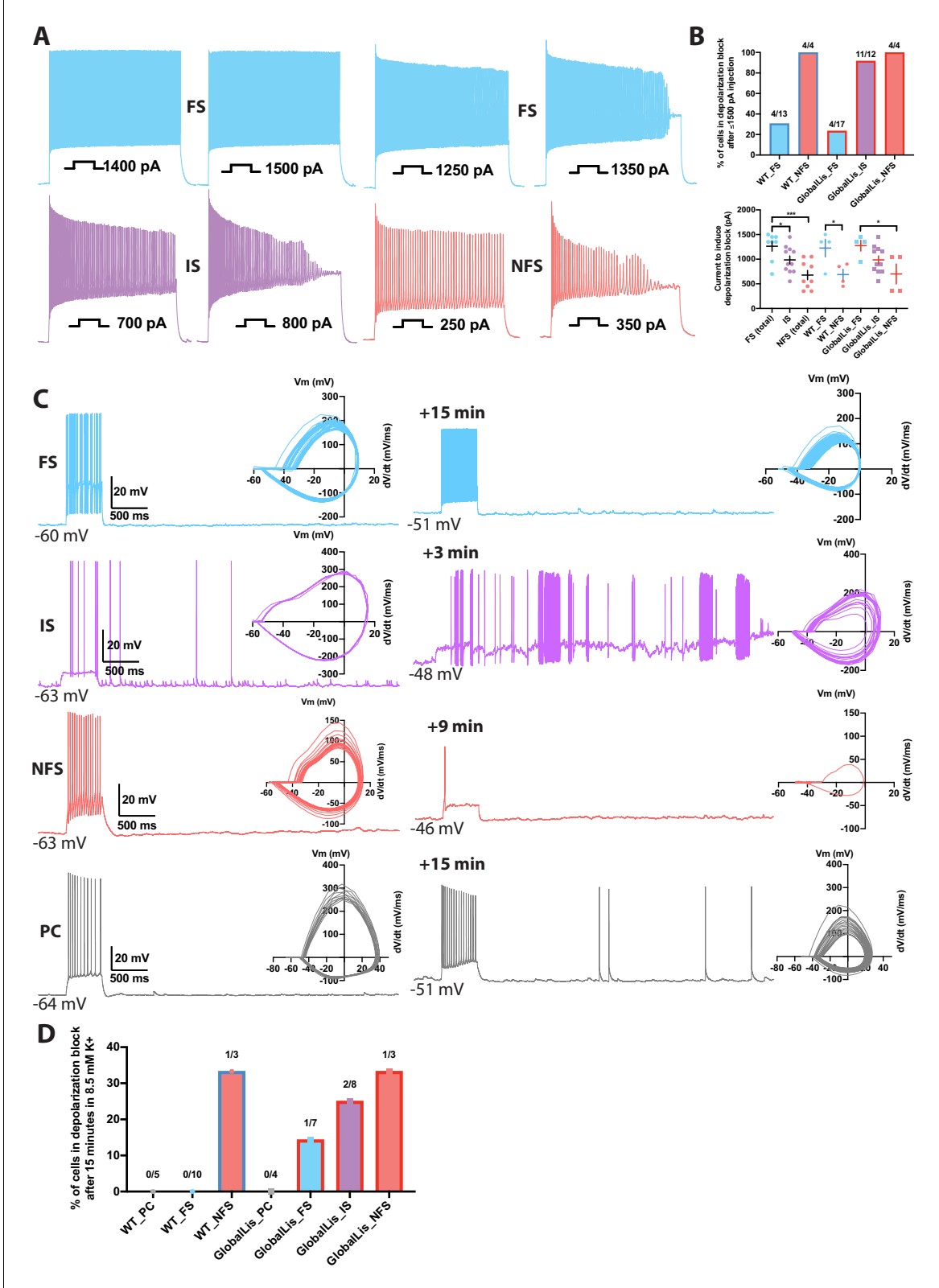

**Figure 6.** Depolarization block in PV+INTs. (**A**) Maximum firing frequencies and depolarizing block in subtypes of PV+INTs. Top left shows FS cell which showed no reduction in firing frequency after current injection of 1500 pA. Top right displays FS cell, bottom left displays IS cell, and bottom right displays NFS cell, all of which blocked. (**B**) Percentages of PV+INT subtypes that went into depolarization block with current injection of 1500 pA or less. For statistical analysis *p<0.05, **p<0.01, ***p<0.001. (**C**) Traces of PV+INTs in 3.5 mM extracellular [K+] (left) and after 8.5 mM extracellular [K+] (right)

*Figure 6 continued on next page*

*Figure 6 continued*

with inset phase plots. The IS cell example (second from top) shows the cell transitioning into depolarizing block and losing the ability to initiate action potentials. The NFS cell example (third from top) trace in 8.5 mM K+ shows the last action potential this cell was able fire before completely entering depolarization block. (D) Left side displays percentages of PCs and PV cells in depolarization block after 15 min or less in 8.5 mM extracellular [K+]. Right side displays percentages of PV+INT physiological subtypes.

The online version of this article includes the following source data for figure 6:

**Source data 1.** Current inducing depolarization block.

to enable targeting of single PV+INT nuclei. We microdissected hippocampi of p20 WT and Global-Lis mice, lysed the cells, and performed snRNA-seq on fluorescence-sorted GFP+ PV+INT nuclei (*Figure 7A*). As a first pass, to establish the identities of the PV+ subtypes, we integrated and aligned this dataset with other established single-cell RNAseq profiles of (i) a publicly available pan-GABAergic Allen Brain Institute mouse dataset (*Tasic et al., 2018*) and (ii) *Nkx2.1*-cre, MGE-derived cortical and hippocampal interneurons (*Mahadevan and Mitra, 2020*) using Seurat v3 (*Butler et al., 2018*; *Stuart et al., 2019*). We assigned the putative identities of *Gad1+ Pvalb+* subtypes as fast-spiking basket cells (BC), axo-axonic cells (AAC) or bistratified (BiC) subtypes, using the marker expressions *Tac1*, *Pthlh* or *Sst* respectively, and these marker expressions mapped well with the reference datasets indicating a high confidence in the quality of snRNAseq (*Figure 7—figure supplements 1–2*). Focusing on the PV+INTs from WT and GlobalLis datasets (1781 and 1623 single-nuclei, respectively) for subsequent analysis, we first observed that the entirety of the UMAP space aligns well between WT and GlobalLis (*Figure 7B$_i$*), and the cell recovery numbers of the BC, AAC and BiC subtypes matches well between the genotypes (*Figure 7B$_{ii}$*). This indicates that *Pafah1b1* haploinsufficiency does not lead to gross differences in the fundamental transcriptomic identities or the overall cell recoveries of PV+INT subtypes (*Figure 7B*).

Because we observed robust disruption in the morphophysiological development and laminar positioning of GlobalLis PV+INTs, we examined the full range of transcriptional impairments triggered by *Pafah1b1* haploinsufficiency in PV+INT subtypes, by performing differential gene expression testing. At a stringent false-discovery rate (FDR) < 0.01, 376 genes passed the 10%-foldchange (FC) threshold across the PV+INT subtypes of which, 126 genes were commonly differentially expressed (DE) between the subtypes and the remaining 250 genes were uniquely DE across the subtypes (*Figure 7C*; *Supplementary file 1*). To assess the broad biological impact of the DE genes (DEG), we applied the Ingenuity Pathway Analysis (IPA) framework. These analyses revealed that the DEGs primarily serve to regulate synaptogenesis signaling pathways, glutamatergic/GABAergic neurotransmission, different distinct guidance cues, cell-cell adhesion, and maintenance of extracellular matrix (ECM; *Figure 7D*; *Supplementary file 2*).

First, we observed misregulated expressions of cell-adhesion molecules (CAMs) belonging to cadherin family (*Cdh8-13*), contactin and related family (*Cntn3-6, Cntnap3,5b*), IgCAM family (*Alcam, Dscam, Dscaml1, Ncam2, Kirrel1,3, Igsf11*), including multiple CAM-modifiers (*St3gal4, St6gal1*) (*Figure 7Ei–iii*). It is notable that multiple members of the CAM family are established regulators of interneuron synapse assembly, axonal and dendritic arborization (*Brennaman and Maness, 2008*; *Brennaman et al., 2013*; *Guan and Maness, 2010*; *Gómez-Climent et al., 2011*; *Kröcher et al., 2014*; *Gao et al., 2018*). Next, we also observed several members of signaling pathways belonging to netrin family (*Ntng1, Unc5b, Dcc, Ntn4*), ephrin family (*Efna5, Epha3, Epha4*), robo family (*Slit1, Slit3*) and semaphorin family (*Sema5a, Sema6a, Neto1, Neto2*), are robustly downregulated subsequent to *Pafah1b1* haploinsufficiency (*Figure 7F$_i$*), many of which have well-defined functions in MGE-derived interneuron migration and morphological development (*Andrews et al., 2006*; *Andrews et al., 2008*; *Tran et al., 2007*; *van den Berghe et al., 2013*; *Steinecke et al., 2014*). Interestingly, a small subset of the DEGs are established to exist in a genetic and biochemical complex with *Pafah1b1* (*Figure 7F$_{ii}$*; *Supplementary file 3*) during the regulation of interneuron development (*Assadi et al., 2003*; *Zhang et al., 2009*; *Livnat et al., 2010*), indicating that these molecules could serve as high-confidence targets for future examination. This collectively indicates that multiple cellular mechanisms can converge toward regulating PV+INT morphology in a *Pafah1b1*-dependent cell-autonomous manner.

Based on the *Pafah1b1* haploinsufficiency associated changes in PV+INT firing properties and propensity for depolarization block, we further scrutinized our transcriptome datasets for cell-

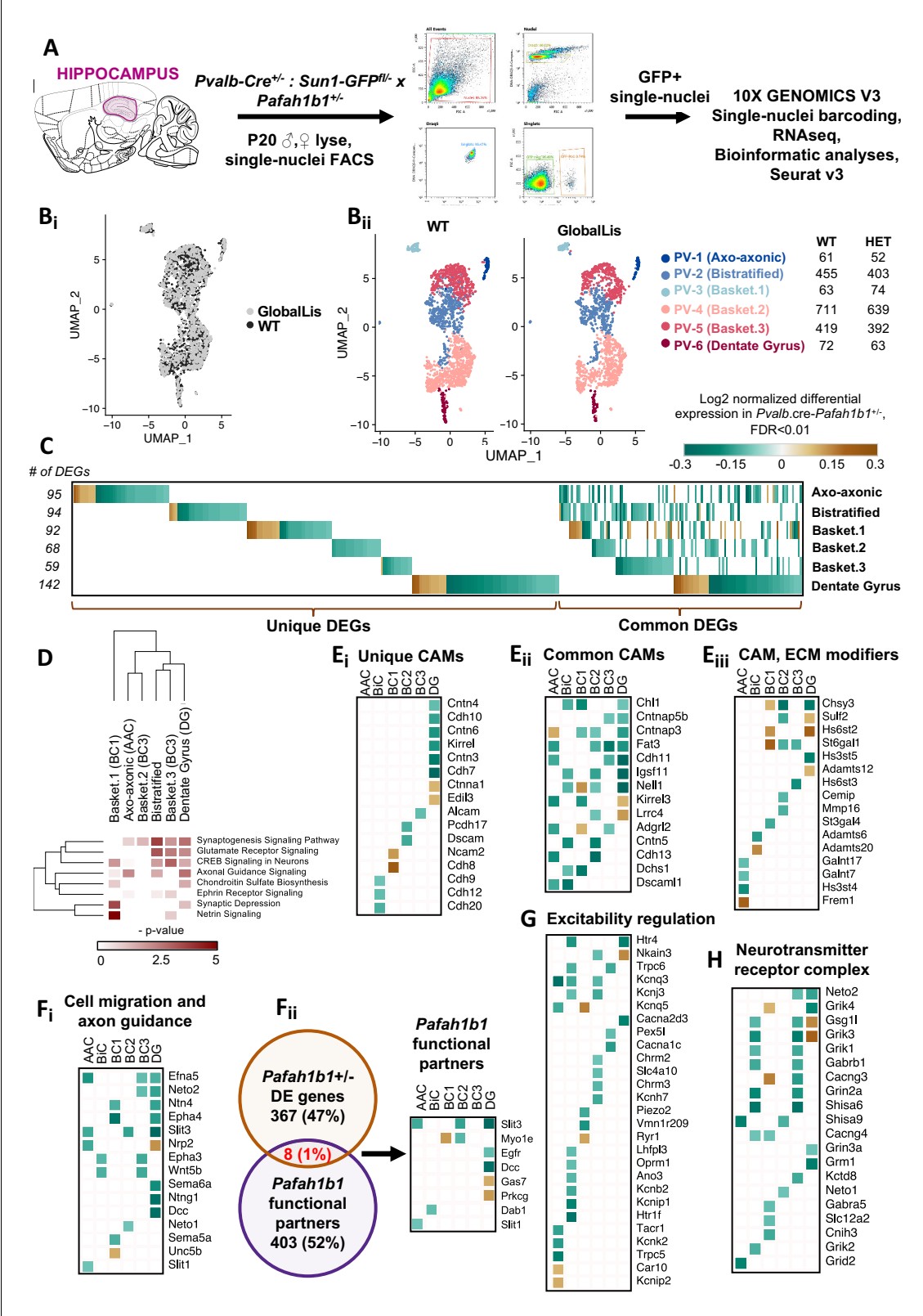

**Figure 7.** Examination of Pafah1b haploinsufficiency on PV+INTs using single-nucleus RNA-seq. (**A**) Overview of the experimental workflow. (**Bi**) Uniform Manifold Approximation and Projection (UMAP) dimensional reduction of single-nuclei transcriptomes of hippocampal PV+INTs, highlighting similar enrichments of the clusters between genotypes. (**Bii**) UMAP visualization of Axo-axonic, Bistratified and Fast-spiking PV+INT subtypes, and table indicating the number of Gad1/Pvalb+ cells recovered in each PV+INT. Cell clusters were color coded and annotated post hoc based on their

*Figure 7 continued on next page*

Figure 7 continued

transcriptional profile identities (Abbreviations: FS, Fast-spiking; DG, Dentate gyrus). (C) Combined heatmap representing the 376 differentially expressed (DE) in hippocampal PV+INTs upon *Pafah1b1* haploinsufficiency, at FDR < 0.01 and Fold Change (FC) >10%, as determined by MAST analysis. (D) Ingenuity Pathway Analysis of significantly overrepresented molecular pathways in each PV+INT subtype. (E–H) Heatmap of log2 FC of significant DE genes in each PV+INT subtype, showing a subset of (Ei) uniquely DE cell-adhesion molecules (CAMs), (Eii) commonly DE CAMs, Eiii, DE extracellular matrix modifying genes; (Fi), genes regulating neuronal migration and axon guidance, (Fii), genes that exist in a genetic and biochemical complex with *Pafah1b1*. (G) regulators of neuronal excitability, and (H) postsynaptic glutamate receptor subunits and associated auxiliary subunits. The online version of this article includes the following figure supplement(s) for figure 7:

**Figure supplement 1.** Integrated analyses of single-cell/nucleus transcriptomes from PV+INTs from Pafah1b1$^{+/+}$, Pafah1b1$^{+/-}$ and reference datasets.
**Figure supplement 2.** Annotation of PV+INT subtypes based on discrete marker gene expressions.

autonomous changes that could impact membrane excitability. Several critical regulators of PV+INT excitability (*Pelkey et al., 2017*) including members of potassium channel family (*Kcnj3, Kcnq5, Kcnh7, Kcnb2, Kcnk2, Kcnip1, Kcnip2*) were misregulated in GlobalLis PV+INTs (*Figure 7G*; *Supplementary file 1*). In addition, two GlobalLis PV+INT clusters displayed altered levels of the key neurotransmitter release regulator *Syt2* (*Supplementary file 1*), potentially reflecting the increased contribution of IS PV+INTs which display reduced unitary amplitudes and release probability in combination with longer latencies. We also observe that the cation/chloride co-transporter NKCC1 (*Slc12a2*) appears to be decreased in subset of basket cells which might also influence somatic chloride extrusion during high-frequency firing in PV+INTs, thereby influencing PV+INT excitability (*Otsu et al., 2020*). Finally, we observed altered expression patterns of glutamate receptors and associated auxiliary subunits (*Grik1-4, Neto1, Neto2*) (*Figure 7H*) that are previously established to regulate excitatory recruitment of PV+INTs (*Pelkey et al., 2017*; *Christensen et al., 2004*; *Wyeth et al., 2017*; *Mulle et al., 2000*).

## Conclusions

Global *Pafah1b1* heterozygous loss has a dramatic effect on the migration and cellular positioning of both excitatory pyramidal cells and inhibitory interneurons (*Fleck et al., 2000*; *Jones and Baraban, 2009*; *D'Amour et al., 2020*). Here, we observed fracturing of stratum pyramidale into heterotopic bands, reduction of PV+INT density in deep hippocampal layers and increased density in the superficial layers. In addition to studying the effects of global heterozygous *Pafah1b1* loss, we utilized cell-type-specific mutations to selectively remove one copy of *Pafah1b1* from PCs (EmxLis) or from medial ganglionic eminence-derived INTs (NkxLis). The overall CA1 structure in the EmxLis mutation resembled the GlobalLis CA1 (disrupted PC layer formation), while the NkxLis mutation was indistinguishable from WT. In contrast, PV+INT radial migration in both the EmxLis and NkxLis genotypes was disrupted in a similar manner to GlobalLis mutation: decreased density in deep layers and increased density in superficial layers. Therefore, PV+INTs require both cell-autonomous *Pafah1b1*-dependent processes, and non-cell-autonomous interactions with pyramidal neurons in order to achieve successful radial migration.

We found that 80% of WT PV+INs were fast-spiking (FS) cells and the remaining 20% were non-fast-spiking (NFS) cells. Compared to FS cells, NFS cells had lower rheobases, higher input resistances, longer action potential half-widths, and lower firing rates. NFS cells consisted of two morphological subtypes: basket cells and a newly identified subtype we have labeled 'radiatum-targeting' PV+ cells, as their axons are confined to stratum radiatum. FS cells consisted of basket cells, axo-axonic cells, bistratified cells and radiatum-targeting cells.

In the GlobalLis mutant, clear morphological labels are difficult to assign to recovered PV+INTs. Cellular morphology was disrupted, with axon patterns resembling combinations of the common morphological subtypes as well as branching into new patterns that have no resemblance to any WT PV+INT. Despite the morphological disruption, clusters of FS and NFS cells with essentially identical electrical properties as WT cells were detected in the GlobalLis mutant. Furthermore, an additional cluster of PV+ cells emerged. We refer to this population as 'intermediate-spiking' (IS) cells as their intrinsic properties fell between those of FS and NFS cells.

Despite pyramidal cell layer disorganization and disrupted radial migration of PV+INTs in the EmxLis mutant, PV+ cells were able to develop into normal morphophysiological subtypes. EmxLis PV+INs cleanly parsed into two clusters: FS and NFS cells. Surprisingly, neither the fractured banding

of PCs or ectopic positioning of INTs interfered with the ability to form identifiable basket, axo-axonic, bistratified, or radiatum-targeting cells. Similar to GlobalLis mutation, the NkxLis hippocampus was composed of PV+ FS, IS, and NFS cells. Despite normal pyramidal cell layer development, morphological development of NkxLis PV+INTs was disrupted, and clear morphological subtypes could not be resolved.

*Pafah1b1* haploinsuffiency disrupts numerous genetic programs controlling PV+INT migration, morphogenesis, synapse formation and cellular excitability. Interestingly, the emergence of IS cells and alteration of morphology in the GlobalLis mutant did not correlate with any additional PV+INT clusters in the snRNA-seq analysis and we did not detect any differences in 'fast-spiking' phenotype-related ion channels such as the Kv3 family of voltage-gated potassium channels. However, the snRNA-seq presents only a snapshot of the transcriptomic aberrations in PV+INTs due to Global *Pafah1b1* haploinsufficiency. Therefore, the lack of a distinct IS population in the snRNA-seq analysis could imply that alterations of a very small number of genes are responsible for changes, which would not result in any additional clustering. It is possible that the cellular changes observed may arise from epigenetic changes, changes in protein expression levels, or changes in modifications of proteins that would not be detectible in this dataset. A third possibility is that changes during development alter the circuit integration and maturation of PV+INTs and these changes might not be evident in their mature transcriptome. Finally, it is possible that the morphological aberrations in PV+INTs are further shaped by the transcriptomic aberrations in other cell types. Because *Pafah1b1*-disruption in pyramidal cells does not seem to affect PV+INT morphological maturation, future studies could examine *Pafah1b1* disruption exclusive to glial cells to examine such non-autonomous mechanisms of regulation of interneuron functions. For example, glial cells that are also impacted by *Pafah1b1* haploinsufficiency, also secrete several guidance cues that shape the assembly, lamination and morphology of interneurons (*Vallee and Tsai, 2006*; *Yokota et al., 2007*). Therefore, such cell-autonomous and non-autonomous mechanisms could converge to regulate PV+INT morphological maturation.

Although IS cells emerge in the GlobalLis and NkxLis mutants, physiologically typical FS and NFS cells also develop in these mice. A full understanding as to why some PV+INTs develop into FS cells while others become IS cells remains elusive. One factor that may influence PV+INT development is the embryonic birth date and the level of excitatory and/or inhibitory drive (*Donato et al., 2015*). Indeed, analysis of spontaneous excitatory post synaptic currents (sEPSCs) in PV+INTs reveals that WT and GlobalLis FS cells receive a much higher level of spontaneous excitatory input than both IS and NFS cells. It is a possibility that PV precursors destined to become NFS cells require a low level of excitatory input to drive their development, while FS cells require a high level of spontaneous excitatory drive. A former study found that PV+INTs in visual cortex are composed of various sub-clusters of cells, and the cluster with the lowest sEPSC frequencies had low firing rates, high input resistances, low rheobases, and wide action potential half-widths, reminiscent of PV+NFS cells (*Helm et al., 2013*). Another study found that overexpressing synapse-associated protein 97 (SAP97) in PV+INTs increases sEPSC frequency, with a corresponding increase in firing rate and decrease in AP half-width (*Akgul and Wollmuth, 2013*). It is possible that IS cells are PV precursors which failed to develop into FS cells due to insufficient excitatory recruitment.

Another possibility considers that cells produce variable amounts of proteins and protein levels change in response to intracellular and extracellular signals (*Dörrbaum et al., 2018*). Cells that either synthesize lower amounts of *Pafah1b1* protein and/or have more disruptions to *Pafah1b1* turnover due to internal and external environmental interactions are expected to have more difficulty with cellular migration and protein trafficking and could consequently develop into IS instead of FS cells. Additionally, or alternatively, the different environments encountered by migrating early-born or late-born PV+INTs could interact with *Pafah1b1* intracellular signaling pathways to differentially affect development of these populations of cells. As the present study did not investigate the role of cellular birth-dating on PV+INT development, future experiments could examine this using BrdU fate mapping in combination with immunohistochemistry and with other techniques that allow labeling of isochronic cells for electrophysiological recording.

We observed that both WT and EmxLis CA1 are composed of 79–80% FS cells (preferentially found in deep layers) and 19–20% NFS cells (preferentially in superficial layers). In contrast, FS cells make up only 44% of GlobalLis and 29% of NkxLis PV+INTs, with the majority (56–71%) consisting of IS/NFS (which are also preferentially found in superficial layers). Compared to FS cells, IS and NFS

cells have lower firing rates and provide less potent, less reliable inhibitory output to pyramidal cells. Thus, the expansion of these non-canonical cell types will erode the precise inhibition usually provided by FS cells, placing network excitability on the precipice for electrographic seizure activity. Finally, due to their susceptibility to depolarization block, IS and NFS cells will likely lose the ability to initiate action potentials and control runaway excitation during seizure episodes, allowing epileptic activity to spread from the hippocampus to other brain regions. Until now, the disrupted neuronal migration in classical lissencephaly has been paradigmatically linked to the generation of epileptiform activity; however, our EmxLis and NkxLis experiments challenge this assumption. We propose that it is not disrupted neuronal migration perse that underlies the origin of epilepsy, but rather disruptions to intrinsic developmental programs that result in the emergence of a less efficient PV+INT cell type.

## Materials and methods

### Animals

All experiments were conducted in accordance with animal protocols approved by the National Institutes of Health. *Pafah1b1*$^{+/fl}$ male mice (provided by Anthony Wynshaw-Boris, Case Western Reserve University) were crossed with Sox2-Cre female mice (provided by National Human Genome Research Institute transgenic core, Tg(Sox2-Cre)1Amc/J). Sox2-Cre females exhibit Cre-recombinase activity in gamete tissues, which allow for genotyping and selection of non-conditional *Pafah1b1*$^{+/-}$ mutants without the Cre allele in a single cross. To identify mutant offspring, we designed a new forward primer (Recombined forward: AGTGCTGGGACAGAAACTCC, Reverse: CCTCTACCACTAAAGCTTGTTC) from the previously published genomic sequences. These mice were bred to wild-type C57BL/6J mice (Jackson Labs stock no. 00064) to maintain global *Pafah1b1*$^{+/-}$ colonies. To obtain cell-type-specific *Pafah1b1* mutations, we crossed *Pafah1b1*$^{+/fl}$ mice to Nkx2.1-Cre (Jackson Labs stock no. 008661, C57BL/6J-Tg(Nkx2-1-cre)2Sand/J) and Emx1-Cre (Jackson Labs stock no. 005628, B6.129S2-Emx1$^{tm1(cre)Krj}$/J) lines.

To enable genetic access and targeting of PV+ cells, *Pafah1b1* mutant lines were crossed to PV-tdTomato (widely referred to as Ai9) reporters. *Pafah1b1*$^{+/-}$ mice were crossed to PV-Cre (Jackson Labs stock no. 017320, B6.129P2-Pvalb$^{tm1(cre)Arbr}$/J) and tdTomato mice (Jackson Labs stock no. 007909, B6.Cg-Gt(ROSA)26Sor$^{tm9(CAG-tdTomato)Hze}$/J). *Pafah1b1*$^{+/fl}$::Nkx2.1Cre and *Pafah1b1*$^{+/fl}$::Emx1Cre lines were crossed to Cre-independent PV-TdTom mice (Jackson Labs stock no. 027395, Tg(Pvalb-tdTomato15Gfng)). For single-cell nuclear RNAseq experiments, *Pafah1b1*+/-::PV-Cre mice were crossed to Sun1-GFP mice (Jackson Labs stock no. 030952, B6.129-Gt(ROSA)26Sor$^{tm5(CAG-Sun1/sfGFP)Nat}$/MmbeJ).

Male and female mice from p19-p60 were used. Mice were housed and bred in a conventional vivarium with standard laboratory chow and water in standard animal cages under a 12 hr circadian cycle.

### Immunohistochemistry on perfused tissue

All IHC experiments were performed in dorsal hippocampus. Mice were deeply anesthetized, and tissue was fixed via transcardial perfusion with 30 mL of phosphate-buffered saline (PBS) followed by 50 mL of 4% paraformaldehyde (PFA) in 0.1 M phosphate buffer (PB, pH 7.6). Brains were post-fixed overnight at 4°C when processed for immunostaining for PV and NeuN. Brains were cryopreserved in 30% sucrose and sectioned on a freezing microtome at 50 µm. Sections were rinsed in PB, blocked for 2 hr in 10% normal goat serum with 0.5% Triton X-100, and then incubated in primary antibody for 2 hr at room temperature or overnight at 4°C. Sections were then rinsed with PB and incubated in secondary antibodies (1:1000) and DAPI (1:2000) for 2 hr at room temperature. All antibodies were diluted in carrier solution consisting of PB with 1% BSA, 1% normal goat serum, and 0.5% Triton X-100. Sections were then rinsed, mounted on Superfrost glass slides, and coverslipped using Mowiol mounting medium and 1.5 mm cover glasses.

### Image acquisition and analysis

Confocal images were taken using a Zeiss 780 confocal microscope. For all slices with immunostained or genetically reported somatic signal, 50-µm-thin sections were imaged using a Nikon

spinning disk (Yokogawa CSU-X) confocal microscope. Counting was performed on four hippocampal sections from each animal. Quantitative analysis of PV+ cell density in each CA1 layer was performed using ImageJ software (NIH, Bethesda, MD, USA).

## Slice preparation

Mice (p19-p60) were anesthetized with isoflurane and then decapitated. The brain was dissected out in ice-cold sucrose artificial cerebrospinal fluid (aCSF) containing the following (in mM): 130 NaCl, 3.5 KCl, 24 NaHCO3, 1.25 $NaH_2PO_4$, 1.5 $MgCl_2$, 2.5 $CaCl_2$, and 10 glucose, saturated with 95% $O_2$% and 5% $CO_2$. Mice older than p30 were dissected in sucrose-substituted artificial cerebrospinal fluid (SSaCSF) containing the following (in mM): 90 sucrose, 80 NaCl, 3.5 KCl, 24 $NaHCO_3$, 1.25 $NaH_2PO_4$, 4.5 MgCl, 0.5 $CaCl_2$, and 10 glucose, saturated with 95% $O_2$% and 5% $CO_2$. Coronal hippocampal slices (300 µm) were cut using a VT-1200S vibratome (Leica Microsystems) and incubated submerged in the above solution at 32–34°C for 30 min and then maintained at room temperature until use. Slices were incubated for at least 45 min before conducting electrophysiological recordings.

## Whole-cell electrophysiology

All recordings were performed in dorsal hippocampus. For patch-clamp recordings following recovery slices were transferred to an upright microscope (Zeiss Axioskop), perfused with aCSF (with or without SCZD as indicated) at 2–3 ml/min at a temperature of 32–34°C. Individual cells were visualized using a 40x objective using fluorescence and IR-DIC video microscopy. Electrodes were pulled from borosilicate glass (World Precision Instruments) to a resistance of 3–5 MΩ using a vertical pipette puller (Narishige, PP-830). Whole-cell patch-clamp recordings were made using a Multiclamp 700B amplifier (Molecular Devices), and signals were digitized at 20 kHz (Digidata 1440A, filtered at 3 kHz) for collection on a Windows computer equipped with pClamp 10.4 software (Molecular Devices). Uncompensated series resistance ranged from 10 to 35 MΩ and was monitored continuously throughout recordings with −5 mV voltage steps. Pipette capacitance compensation and bridge balance were applied in current-clamp experiments. For current-clamp and voltage-clamp recordings of PV-TdTom+ interneurons and CA1 pyramidal cells, two different internal solutions containing (in mM) were used: (A) 130 K-gluconate, 5 KCl, 10 HEPES, 3 MgCl2, 2 Na2ATP, 0.3 NaGTP, 0.6 EGTA, and 0.2% biocytin (calculated chloride reversal potential ($E_{Cl}$) of −67 mV) or (B) 130 K-Gluconate, 10 KCl, 10 HEPES, 3 MgCl2, 2 Na2ATP, 0.3 NaGTP, 0.6 EGTA, and .2% biocytin (calculated $E_{Cl^-}$ = −27 mV).

Input resistance (Rin) was measured using a linear regression of voltage deflections in response to 500 ms-long current steps of four to six different amplitudes (−200 to +50 pA, increments of 50 pA). Apparent membrane time constant (τm) was calculated from the mean responses to 20 successive hyperpolarizing current pulses (−20 pA; 400 ms) and was determined by fitting voltage responses with a single exponential function. Action potential (AP) threshold was defined as the voltage at which the slope trajectory reaches 10 mV/ms. AP amplitude was defined as the difference in membrane potential between threshold and the peak. AP half-width was measured at the voltage corresponding to half of the AP amplitude. Afterhyperpolarization (AHP) amplitude was defined as the difference between action potential threshold and the most negative membrane potential attained during the AHP. These properties were measured for the first action potential elicited by a depolarizing 500 ms-long current pulse of amplitude just sufficient to bring the cell to threshold for AP generation (rheobase). The adaptation ratio was defined as the ratio of the average of the last three interspike intervals relative to the first three interspike intervals during a 500 ms-long spike train elicited using twice the rheobase. Firing patterns were investigated by a series of 500 to 800 ms-long current injections (step size 50 pA) until 3x threshold current was reached or depolarization-block was induced. Firing frequency was calculated from the number of spikes observed during the first 500 ms of the spike train. Firing frequency at 3x threshold was substituted by maximum firing frequency in cells with depolarization-block. Ihyp Sag index of each cell was determined by a series of 500 ms-long negative current steps to create V-I plots of the peak negative voltage deflection (Vhyp) and the steady-state voltage deflection (average voltage over the last 200 ms of the current step; Vsag) and used the ratio of (Vrest−Vsag)/(Vrest−Vhyp) for current injections corresponding closest to Vsag=−80 mV.

Spontaneous excitatory post synaptic currents (sEPSCs) were recorded from PV+INTs over continuous sweeps of 10–30 s. At least 50 events were sampled to determine sEPSC frequency, these events were averaged to determine average sEPSC amplitude and apparent sEPSC decay time constant. For PV+INT-CA1PC paired recordings, the presynaptic PV+INT was held in current clamp with membrane potential biased to −70 mV, while the postsynaptic cell was held voltage clamp at −70 mV. Synaptic transmission was monitored by producing action potentials in presynaptic PV+INTs (held in current-clamp around −70 mV) every 10 s by giving 2 ms 1–2 nA current steps. Presynaptic trains to probe unitary transmission dynamics consisted of 25 presynaptic action potentials at 50 Hz. Basal unitary event properties for each cell were analyzed using 10 consecutive events obtained 4–5 min after establishing the postsynaptic whole-cell configuration. Amplitudes reflect the average peak amplitude of all events including failures, potency is the average peak amplitude excluding failures. Decay kinetics were measured by single exponential fit of uIPSC potency. The latency of synaptic transmission was defined as the time from the peak of the AP to 5% of the uIPSC potency.

For depolarization block experiments, WT and GlobalLis TdT+PV+INTs were recorded in current-clamp mode. Rheobase, input resistance, AP half-width, firing frequency at two x threshold x and maximum firing frequency were recorded (and used to identify physiological subtype) using above protocols. Further depolarizing current was injected in 500 ms sweeps with a step size of 50 pA until cells were subject to depolarizing block or 1500 pA of current were injected. Following this, 10 s sweeps of spontaneous activity with 500 ms of evoked firing (with ~1.5 x threshold current) were taken. After obtaining a stable baseline (~10 sweeps), the extracellular solution was switched to a solution with 8.5 mM [K+] the same 10 s (500 ms evoked firing) sweeps were taken for up to 15 min.

## Principle component analysis

The optimal number of clusters were computed by the published NbClust package in R based on Euclidean distances of normalized (log transformed) intrinsic electrophysiological parameters (*Charrad et al., 2014*; R-studio version 0.99.451 and R version 3.4.2.). After determining the optimal number of clusters, principal components analysis (PCA) and hierarchical clustering of normalized intrinsic electrophysiological properties based on Euclidean distance were performed using K-Means analysis. To confirm accuracy of K-means clustering, a separate algorithm was performed using Ward's Test (again using R) and the resulting clusters were compared to the original K-means clusters.

## Anatomical reconstructions and morphological analysis

After biocytin filling during whole-cell recordings, slices were fixed with 4% paraformaldehyde and stored at 4℃ then permeabilized with 0.3% Triton X-100 and incubated with Alexa Fluor 488 or Alexa Fluor 555-conjugated streptavidin. Resectioned slices (75 μm) were mounted on gelatin-coated slides using Mowiol mounting medium. Cells were visualized using epifluorescence microscopy and images for representative examples were obtained with confocal microscopy. Cells were reconstructed and analyzed with Sholl analysis using Neurolucida software (MBF Bioscience). Polar histograms of WT and GlobalLis PV+INT dendrites and axons were created using the Neurolucida function (10 degree bins). Polarity preference was determined by calculating the percentage of horizontally (150–210, 330–30 degrees) or vertically (60–120, 240–300 degrees) oriented axon in each genotype.

## Statistical analysis

All data were first tested for normality and then tested with unpaired t-tests, Mann-Whitney tests, One-Way ANOVA with Holm-Sidak multiple comparisons tests, Kruskal-Wallis tests with Dunn's multiple comparison tests, or Fisher exact tests as appropriate (Graphpad Prism). Quantification and error bars display standard error of the mean. Intrinsic electrophysiological parameters in the text are values for the upper and lower 95% confidence intervals of the mean. Values shown for unitary synaptic transmission properties consist of mean and standard error of the mean.

## Single nucleus isolation

Hippocampus from seven mutant (*Pafah1b1$^{+/-}$;PV-Cre$^{+/-}$;Sun1-GFP$^{+/-}$*) and six WT (*Pafah1b1$^{+/+}$; PV-Cre$^{+/-}$; Sun1-GFP$^{+/-}$*) P21 mice were quickly dissected in ice-cold DPBS, immediately frozen on dry

ice and stored at −80℃. We pooled mutant or WT hippocampus into a Dounce Homogenizer containing 1 mL freshly prepared ice-cold lysis buffer (low sucrose buffer with 1 mM DTT, 0.1% NP-40), applying 10 strokes with pestle A followed by 10 strokes with the pestle B. The homogenate was filtered through a 40-µm cell strainer, transferred to a DNA low bind 2 mL microfuge tube and centrifuged at 300 g for 5 min at 4℃. The supernatant was removed, the pellet was gently resuspended in a low-sucrose buffer (320 mM sucrose, 10 mM HEPES-pH 8.0, 5 mM $CaCl_2$, 3 mM Mg-acetate, 0.1 mM EDTA) and centrifuged for another 5 min. The nuclei were resuspended in 500 µl 1xPBS with 1% BSA and 0.2 U/µl SUPERaseIn RNase Inhibitor (ThermoFisher, #AM2696) and loaded on top of 900 µl 1.8 M Sucrose Cushion Solution (Sigma, NUC-201). The sucrose gradient was centrifuged at 13,000 g for 45 min at 4℃. The supernatant was discarded, and the nuclei were resuspended in 500 ul Pre-FACS buffer (1xPBS with 1% BSA, 0.2 U/µl SUPERaseIn RNase Inhibitor and 0.2 M sucrose). Before sorting, nucleus from the six WT or seven mutant mice were pooled together and 5 µl of 5 mM DRAQ5 were added.

Samples were processed on a Sony SH800 Cell Sorter with a 100 mm sorting chip. 15,000 GFP+/DRAQ5+ nuclei from mutant and WT samples were collected directly into 1.5 ml centrifuge tubes containing 10 µl of the Pre-FACS buffer. PCR cycles were conducted for cDNA amplification, and the subsequent library preparation and sequencing were carried out in accordance with the manufacturer recommendation (Chromium Single Cell 3' Library and Gel Bead Kit 10X v3, 16 reactions). Sequencing of the libraries were performed on the Illumina HiSeq2500 at the NICHD, Molecular Genomics Core facility. The cell number estimates, mean reads per cell (raw), median genes per cell respectively, are as follows $Pafah1b1^{+/-}$: 8470, 22,289, 2024; $Pafah1b1^{+/-}$: 8185, 24,652, 2142. Demultiplexed samples were aligned to the mouse reference genome (mm10). The end definitions of genes were extended 4 k bp downstream (or halfway to the next feature if closer) and converted to mRNA counts using the Cell Ranger Version 2.1.1, provided by the manufacturer.

## RNAseq data processing and analyses, differential expression testing, and visualization

Processing (load, align, merge, cluster, differential expression testing) and visualization of the scRNAseq datasets were performed with the R statistical programming environment (v3.5.1) (*R Development Core Team, 2013*), and Seurat package (v3.1.5) (*Butler et al., 2018*; *Stuart et al., 2019*). The $Pafah1b1^{+/+}$ and $Pafah1b1^{+/-}$ datasets were first merged with the Allen Institute reference and in-house MGE interneuron reference datasets. To analyze the Allen Institute mouse dataset of the single-cell transcriptomes of ~76,000 cells from >20 areas of mouse cortex and hippocampus, we downloaded the transcriptome/HDF5 file (https://portal.brain-map.org/atlases-and-data/rnaseq) and subsequently converted into Seurat v3-compatible format based on the instructions provided in the Allen Institute Portal (https://portal.brain-map.org/atlases-and-data/rnaseq/protocols-mouse-cortex-and-hippocampus) and custom scripts in R package as previously described (*Chittajallu et al., 2020*). Single-cell transcriptomes from *Nkx2.1*-cre:Ai14, MGE-derived cortical and hippocampal interneurons (postnatal day 18–20) were processed as previously described (*Mahadevan and Mitra, 2020*). To perform integrated analyses, we identified a common set of genes between $Pafah1b1^{+/+}$, $Pafah1b1^{+/-}$, *Nkx2.1*-MGE cortical and hippocampal interneurons and Allen datasets, and utilized these for the initial analyses in *Figure 7—figure supplements 1–2*. Data set preprocessing, normalization, identification of variable genes, canonical correlation analyses were performed according to default Seurat parameters, unless otherwise mentioned. Quality control filtering was performed by only including cells that had between 200 and 20000 unique genes, and that had <5% of reads from mitochondrial genes. Clustering was performed on the top 25 PCs using the function FindClusters() by applying the shared nearest neighbor modularity optimization with clustering resolution of 0.5. Phylogenetic tree relating the 'average' cell from each identity class based on a distance matrix constructed in gene expression space using the BuildClusterTree() function. Overall, we identified 36 clusters using this approach, among which clusters 12, 13, 16, 17 are highly enriched in the $Pafah1b1^{+/+}$, $Pafah1b1^{+/-}$ datasets and aligned well in their corresponding UMAP spaces with the reference datasets. The identities of clusters 12, 13, 16, 17 are matched with the top gene markers identified by the FindAllMarkers(). These four clusters are *Gad1*+, *Pvalb*+ and *Vip*-, and were putatively annotated as Bistratified, Fast-spiking, and Axo-axonic subsets of PV+INTs, based on marker expression of *Sst*, *Tac1*, and *Pthlh*, respectively, as indicated in interneuron literature and previous scRNAseq studies (*Fishell and Kepecs, 2020*; *Hodge et al., 2019*; *Paul et al.,*

*2017*; *Pelkey et al., 2017*; *Saunders et al., 2018*; *Tasic et al., 2016*; *Tasic et al., 2018*; *Yao et al., 2020*; *Harris et al., 2018*). Subsequent to dataset validation using references, the *Pafah1b1*$^{+/+}$ and *Pafah1b1*$^{+/-}$ datasets reanalyzed by subsetting the cells that expressed *Gad1*, *Pvalb* expression >0.1 and by excluding the cells containing non-PV+INT genes *Slc17a7*, *Ttr*, *Scn3a*, *Gpc5*, *Slc1a2*, *Htr2c*, *Trpm3* expressions < 0.1. Clustering was performed on the top 25 PCs using the function FindClusters() by applying the shared nearest neighbor modularity optimization with clustering resolution of 0.5. Similar to prior analyses, we recovered Bistratified, Fast-spiking and Axo-axonic subsets of PV +INTs. Additionally, a minor population of *Nos1*+ cells clearly segregated, which represents putative dentate gyrus-expressed PV+INTs (*Vaden et al., 2020*; *Shen et al., 2019*; *Jinno and Kosaka, 2002*).

Differential gene expression testing was performed using the MAST package within the FindMarkers() function to identify the differentially expressed genes between two subclusters (*Finak et al., 2015*). We applied a stringent false-discovery rate <0.01, and minimum logFC in our DEGs as ±0.1, because MAST has been previously reported to underestimate the magnitude of fold change (*Ximerakis et al., 2019*, *Mahadevan and Mitra, 2020*). Moreover, previous studies have demonstrated the MAST approach for DEG testing to be powerful in determining subtle changes in highly transcribed genes, and among abundant populations, additional to underrepresenting changes among weakly transcribed genes (*Finak et al., 2015*; *Ximerakis et al., 2019*). Molecular and functional annotation of the DEGs were conducted using Ingenuity Pathway Analyses (IPA) platform, to identify the biological pathways and disease pathways over-represented. Experimentally validated and predicted *Pafah1b1* genetic and protein interactions were obtained via publicly available data mining resources (*Kotlyar et al., 2016*; *Rahmati et al., 2017*) IPA was also used to annotate genes with their known cellular functional classes. Heatmaps for the DEGs were generated using the Morpheus package (https://software.broadinstitute.org/morpheus) within the R framework.

## Acknowledgements

We thank Dr. Carolina Bengtsson-Gonzales for developing the code for cluster analysis. We thank Drs. Kenneth Pelkey and Ramesh Chittajallu for discussing project ideas and technical details of experiments. We thank Steven Hunt for genotyping and cell processing for microscopy. We thank Daniel Abebe for animal support. We thank Dr. Vincent Schram and the NICHD imaging core for confocal microscopy support. We thank Drs. Steven L Coon, Tianwei Li and James R Iben at the Molecular Genomics Core, NICHD, for RNA sequencing and bioinformatics support. We thank Drs. Apratim Mitra and Ryan Dale (NICHD Bioinformatics and Scientific Programming Core) for assistance with integrated analysis using the Allen Brain scRNAseq dataset. These analyses utilized the computational resources of the NIH HPC Biowulf cluster (http://hpc.nih.gov). We thank Dr. Anthony Wynshaw-Boris for providing the heterozygous floxed *Pafah1b1* mouse.

## Additional information

### Funding

| Funder | Grant reference number | Author |
| --- | --- | --- |
| Eunice Kennedy Shriver National Institute of Child Health and Human Development | Intramural Resarch Award | Chris J McBain |

The funders had no role in study design, data collection and interpretation, or the decision to submit the work for publication.

### Author contributions

Tyler G Ekins, Conceptualization, Data curation, Formal analysis, Validation, Investigation, Visualization, Methodology, Writing - original draft, Writing - review and editing; Vivek Mahadevan, Formal analysis, Validation, Investigation, Methodology, Writing - original draft, Writing - review and editing; Yajun Zhang, Investigation, Methodology; James A D'Amour, Conceptualization, Writing - original draft, Writing - review and editing; Gülcan Akgül, Methodology; Timothy J Petros, Resources,

Supervision, Investigation, Methodology; Chris J McBain, Conceptualization, Resources, Supervision, Funding acquisition, Writing - original draft, Project administration, Writing - review and editing

### Author ORCIDs
Tyler G Ekins (iD) https://orcid.org/0000-0002-9801-4843
Vivek Mahadevan (iD) http://orcid.org/0000-0002-0805-827X
James A D'Amour (iD) http://orcid.org/0000-0002-8144-3692
Chris J McBain (iD) https://orcid.org/0000-0002-5909-0157

### Ethics
Animal experimentation: All mouse experiments were conducted in accordance with animal protocols approved by the National Institutes of Health (ASP# 17-045).

### Decision letter and Author response
Decision letter https://doi.org/10.7554/eLife.62373.sa1
Author response https://doi.org/10.7554/eLife.62373.sa2

## Additional files

### Supplementary files
• Supplementary file 1. Transcriptional impairments triggered by Pafah1b1 haploinsufficiency in PV+INT subtypes.

• Supplementary file 2. Ingenuity Pathway Analysis (IPA) analysis of differentially expressed genes.

• Supplementary file 3. Genetic and biochemical interactions with Pafah1b1.

• Transparent reporting form

### Data availability
Data generated are included in the manuscript, supporting files, and source data.

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
