## [Decision Letter]

**Acceptance summary:**

This paper examines the properties of GABAergic interneurons in a mouse model of type I lissencephaly, a neurodevelopmental disorder. Utilizing electrophysiology, immunocytochemistry and RNA sequencing, the authors found that the lissencephaly mutation reduces the abundance of fast-spiking PV+ interneurons, while increasing the proportion of neurons with intermediate spiking phenotype. The mutation changes morphological development, intrinsic excitability, and inhibitory output of PV+ interneurons. Single-cell RNA sequencing reveals several dysregulated genes related to morphogenesis, cell excitability and synapse formation. These results suggest that impaired development and function of PV+ interneurons contributes to the spontaneous seizures observed in type I lissencephaly.

**Decision letter after peer review:**

Thank you for submitting your article "Emergence of Non-Canonical Parvalbumin-Containing Interneurons in Hippocampus of a Murine Model of Type I Lissencephaly" for consideration by *eLife*. Your article has been reviewed by three peer reviewers, and the evaluation has been overseen by a Reviewing Editor and John Huguenard as the Senior Editor. The following individual involved in review of your submission has agreed to reveal their identity: Gordon Fishell.

The reviewers have discussed the reviews with one another and the Reviewing Editor has drafted this decision to help you prepare a revised submission.

Summary:

The paper by Ekins et al. examines the properties of GABAergic interneurons in a mouse model of type I lissencephaly, a neurodevelopmental disorder. To address this question, the authors use electrophysiology, immunocytochemistry, and molecular techniques including RNA seq. The main findings are:

– The lissencephaly mutation reduces the abundance of fast-spiking PV+ interneurons, and increases the proportion of neurons with intermediate spiking phenotype.

– Global and cell-specific mutations in GABAergic interneurons have comparable effects, suggesting cell-autonomous mechanisms.

– The mutation changes morphological development, intrinsic excitability, and inhibitory output of PV+ interneurons.

– Single-cell RNA sequencing reveals several dysregulated genes, related to morphogenesis, cell excitability, and synapse formation.

Based on these results, the authors conclude that impaired development and function of PV+ interneurons contributes to the spontaneous seizures observed in type I lissencephaly. Overall, the reviewers found this a quite nice paper. The key findings are exciting, the experiments are systematically planned and well performed, and the manuscript is carefully written. However, the reviewers have a couple of points that need to be addressed before the manuscript can be published.

Major points:

1) The authors demonstrate that the macroscopic morphological properties of axons and dendrites of PV+ neurons differ between wild type and mutant. However, the microscopic morphological properties are largely ignored. Parameters of interest include dendritic and axonal diameters, number of presynaptic terminals, and "aspinyness" versus "spinyness" of interneuron dendrites.

2) A different clustering analysis must be applied to validate the main conclusions.

The Materials and methods section, regarding clustering analysis, states that: "To identify potential subclusters of PV+INTs, we performed principal components analysis (PCA) and hierarchical clustering based on Euclidean distance of normalized (log transformed) intrinsic electrophysiological parameters using R-studio version 0.99.451 and R version 3.4.2." So the question is, how many clusters exist in the sample? What value should be considered for k? Figure legends seem to provide the answer: "The dendrogram inset indicates 2 optimal clusters" and "The dendrogram inset indicates 3 optimal clusters".

Unfortunately, dendrograms cannot (usually) tell you how many clusters you should have, unless the ultrametric tree inequality holds, which is very rare for any real-world data. In other words, it is a common mistake (and misinterpretation) to use dendrograms as a tool for determining the number of clusters in a dataset. As a result of that premise, one of the central arguments is that "WT PV+INTs consist of two physiological subtypes: FS and NFS cells" whereas "GlobalLis PV+INTs consist of three physiological subtypes: FS, IS and NFS cells". But let us look at the data. Figure 3E and Figure 4E show cell intrinsic properties for these groups. Shouldn't the FS group be similar among these Figures? In fact, the FS groups seems indeed similar if we combine the FS+IS groups in Figure 4. This is because k=2 was assumed in Figure 3 and k=3 was assume in Figure 4, so the FS group suffered further splitting into a new IS subgroup.

3) The authors claim that connectivity of PV+ interneurons and pyramidal neurons differs between wild-type and mutant. To understand the network relevance, it would be nice to know whether this not only holds for inhibitory output, but also for excitatory synaptic input of PV+ interneurons. Mutual connectivity between interneurons might be also important. At the very least, these points need to be better discussed.

4) The conclusions stand and fall with the assumption that recording conditions are the same for wild-type and mutant mice. Controls should be provided to reassure that this is the case, and differences are not dependent on systematic differences in age of the animals, slice quality, etc.

5) The statistics of the paper requires improvement. A Fisher exact test should be used to test the statistical significance of the different proportions between wild-type and mutant mice. "nonparametric t-test" replaced by more specific information, such as Wilcoxon signed rank test or Mann-Whitney U test. Finally, the authors should revise the estimation of the number of classes in the cluster analysis. Based on the shape of the frequency distribution, the authors suggest that two classes in wild-type and three classes in mutant describe the experimental observations. However, the differences in the distributions are quite small, and the significance is unclear.

6) The authors demonstrate several structural and functional differences between PV+ interneurons in wild-type and mutant brains, but what this means for the activity of interneurons in vivo remains unclear. Ideally, the authors should record from PV+ interneurons in wild-type and mutant mice under in vivo conditions (some tetrode recordings might be affordable). At the very least, the results should be better discussed in the context of in vivo activity.

7) The mechanisms underlying the differences in excitability and synaptic output between fast and intermediate spikers remains unclear. Changes in in excitability are likely to be related to changes in Na^+^ or K^+^ channel density or subtype. Similarly, changes in inhibitory output will be due to changes in q, N, or pR. Both are only tangentially related to the RNAseq data reported in the paper. More work seems needed to work out the mechanisms.

8) A bit more emphasis on whether there were any indications in the non-autonomous Emx1-cre mutant would have a made a nice complement (might have missed this if it was there).

9) Another concern is with confusing IS cells with the SST populations, as they are both more or less IS and they apparently had no specific marker to identify the PV cells other than the physiology that was sufficiently perturbed to confound proper identification of this population. That they were discernible seemed evident from the scRNA-seq analysis but some attention to the SST interneurons (which clearly would be affected in the nkx2.1-cre KO) would have been a nice addition. Are they still facilitating, is their morphology normal, Are they properly dendritically targeting. Both distinguishing these for the IS population and describing this population at least a bit more thoroughly would have been very nice additions to what is clearly a wonderful paper.

10) The presentation of the manuscript requires improvement. The amount of description of wild type PV interneurons is surprising, as it has already been thoroughly described, in particular in Pelkey et al., 2017. This is necessary for the comparison of wt and mutant cells but perhaps could be parred down rather than spending the first third of the paper describing things are already well documented in the literature. In the present form, the Discussion is rudimentary and focused on side-issues. Several other aspects need to be included (see other major points).

---

## [Author Response]

Major points:1) The authors demonstrate that the macroscopic morphological properties of axons and dendrites of PV+ neurons differ between wild type and mutant. However, the microscopic morphological properties are largely ignored. Parameters of interest include dendritic and axonal diameters, number of presynaptic terminals, and "aspinyness" versus "spinyness" of interneuron dendrites.

To increase the scope of our macroscopic morphological analysis, we created polar histograms to analyze dendritic and axonal polarity preferences of WT and mutant PV+INTs. We detected a difference in axon orientation preference in Lis PV cells, which markedly differs from the preferred orientations of those in WT PV cells. This information has been added to Figure 2. In addition, we analyzed morphologies of PV+ cells from the EmxLis and NkxLis genotypes. We found a similar pattern among PV+INTs in these genotypes: NFS cells in these genotypes also have less extensive dendrite and axons than EmxLis FS cells and NkxLis FS and IS cells. This is now in the manuscript as Figure 4—figure supplement 3.

We also agree that a thorough examination of the microscopic morphological properties would be useful information, however a proper investigation of these features requires electron microscopy, which is beyond the current scope of this study and our present capabilities.

2) A different clustering analysis must be applied to validate the main conclusions.The Materials and methods section, regarding clustering analysis, states that: "To identify potential subclusters of PV+INTs, we performed principal components analysis (PCA) and hierarchical clustering based on Euclidean distance of normalized (log transformed) intrinsic electrophysiological parameters using R-studio version 0.99.451 and R version 3.4.2." So the question is, how many clusters exist in the sample? What value should be considered for k? Figure legends seem to provide the answer: "The dendrogram inset indicates 2 optimal clusters" and "The dendrogram inset indicates 3 optimal clusters".Unfortunately, dendrograms cannot (usually) tell you how many clusters you should have, unless the ultrametric tree inequality holds, which is very rare for any real-world data. In other words, it is a common mistake (and misinterpretation) to use dendrograms as a tool for determining the number of clusters in a dataset. As a result of that premise, one of the central arguments is that "WT PV+INTs consist of two physiological subtypes: FS and NFS cells" whereas "GlobalLis PV+INTs consist of three physiological subtypes: FS, IS and NFS cells". But let us look at the data. Figure 3E and Figure 4E show cell intrinsic properties for these groups. Shouldn't the FS group be similar among these Figures? In fact, the FS groups seems indeed similar if we combine the FS+IS groups in Figure 4. This is because k=2 was assumed in Figure 3 and k=3 was assume in Figure 4, so the FS group suffered further splitting into a new IS subgroup.

We think there may be some misunderstanding in the methods we used to validate our cluster analysis, however we have extended our analysis to incorporate and compare multiple clustering algorithms. The optimal numbers of clusters were computed by the published NbClust package in R using Euclidean distances (Charrad et al., 2014); this is now stated in the Materials and methods section of the manuscript. We have also added silhouette plots to Figures 3 and 4. To validate the accuracy of the Kmeans clustering, we have also now performed a Ward Test, and then compared results to the original K-means clustering. In the WT and EmxLis genotypes, Ward clustering was identical to the Kmeans clustering. In the GlobalLis and NkxLis genotypes, Ward clustering was also very similar with Kmeans clustering, please see Figure 4—figure supplement 2 for the comparison of these clustering algorithms: the numbers at the bottoms of the column and ends of the row indicate respectively the number of cells in a given K-means or Ward-cluster. The numbers in the table indicate how many cells of a K-means cluster are contained within a given Ward cluster.

Based on the similar sorting by separate clustering algorithms and the consistency and across multiple genotypes, we are confident that the IS cells represent a novel population of PV+INTs that emerge following cell-autonomous *LIS1* mutations within interneurons. Regarding the comment on the identity of FS and IS cells, the clustering algorithm we used indicates that if we were to collapse the number of GlobalLis and NkxLis clusters to artificially select k=2, the IS cells would be part of the NFS cluster, not part of the FS cluster (the resulting dendrograms from the GlobalLis mouse are displayed in Author response image 1).

**Author response image 1. sa2fig1:** 

Sorting using k=2 also results in highly significant differences between intrinsic properties of FS cells and the merged NFS/IS group. Results for select intrinsic properties important for classifying FS/IS/NFS cells both ways are plotted in Author response image 2.

3) The authors claim that connectivity of PV+ interneurons and pyramidal neurons differs between wild-type and mutant. To understand the network relevance, it would be nice to know whether this not only holds for inhibitory output, but also for excitatory synaptic input of PV+ interneurons. Mutual connectivity between interneurons might be also important. At the very least, these points need to be better discussed.

We thank the reviewers for this suggestion and have extended these datasets with new data. In the manuscript, we report that WT FS cells have higher connectivity rates with pyramidal neurons than NFS cells (FS: 20/62 connected, NFS: 3/22 connected; Figure 7C). While this was not statistically significant using a Fisher exact test, the p-value was close to significance (p=.105). This information has been added to the manuscript and a graph is displayed in Author response image 3.

**Author response image 3. sa2fig3:** 

We do not claim that GlobalLis PV+INTs have lower connectivity rates than WT PV+INTs. A graph summarizing WT vs GlobalLis connectivity tested with a Fisher exact test is displayed in Author response image 4.

**Author response image 4. sa2fig4:** 

The suggestion to include data from excitatory inputs onto each cell type is an excellent one. To address excitatory input to PV+INTs, we analyzed spontaneous excitatory post synaptic currents on WT and GlobalLis PV+INTs in and found another pattern that persists across genotypes: FS cells receive a significantly higher frequency of spontaneous excitatory input than both IS (in genotypes that have them) and NFS cells. The summary of spontaneous excitatory post synaptic current frequency, amplitude, and apparent decay time constant are now present in the manuscript and summarized in Figure 5—figure supplement 1. The present study did not focus on PV-PV inhibition, but we agree that it would be an interesting area of future research.

4) The conclusions stand and fall with the assumption that recording conditions are the same for wild-type and mutant mice. Controls should be provided to reassure that this is the case, and differences are not dependent on systematic differences in age of the animals, slice quality, etc.

Quality control (including slice quality) was taken seriously in this project. Male and female mice were used in approximate proportions in this study. We updated Table 1 and Table 2 to report the mean postnatal date at the time of recording (there were no significant differences in age between genotypes or physiological subtypes of cells).

5) The statistics of the paper requires improvement. A Fisher exact test should be used to test the statistical significance of the different proportions between wild-type and mutant mice. "nonparametric t-test" replaced by more specific information, such as Wilcoxon signed rank test or Mann-Whitney U test. Finally, the authors should revise the estimation of the number of classes in the cluster analysis. Based on the shape of the frequency distribution, the authors suggest that two classes in wild-type and three classes in mutant describe the experimental observations. However, the differences in the distributions are quite small, and the significance is unclear.

We added the specific statistical tests used to the Materials and methods section. We also performed a Fisher exact test between the proportions of WT and GlobalLis PV+INT physiological subtypes (p<.0001). The graph is now displayed as Figure 4G and the manuscript and figure legends have been updated.

We disagree with the comment that “…the authors should revise the estimation of the number of classes in the cluster analysis. Based on the shape of the frequency distribution, the authors suggest that two classes in wild-type and three classes in mutant describe the experimental observations. However, the differences in the distributions are quite small, and the significance is unclear.” Please see our response to major point 2 for a discussion of PV+INT cluster analysis.

Regarding significance, Figure 3 displays differences in intrinsic properties of WT FS and NFS cells, many of which (including input resistance, firing frequency at 2x and 3x threshold, adaptation ratio, rheobase, action potential half-width, and sag index) are highly significant. Figures 4 displays this information for GlobalLis, Figure 3—figure supplement 1 for EmxLis and Figure 4—figure supplement 1 for NkxLis PV+INTs, again with many of the same highly significant differences in intrinsic properties between physiological subtypes. Finally, in Figure 4—figure supplement 3 and Figure 4—figure supplement 4 distributions of, respectively, physiological and morphological properties of PV+INTs are shown by physiological subtype and genotype. While we detect almost no changes in these properties of physiological subtypes of PV+INTs based on their genotype, we observed many highly significant differences in intrinsic properties between FS, IS and NFS cells. Please see our response to Major Point 2 for examples of intrinsic physiological properties that are significantly different between FS cells and NFS and/or IS cells.

6) The authors demonstrate several structural and functional differences between PV+ interneurons in wild-type and mutant brains, but what this means for the activity of interneurons in vivo remains unclear. Ideally, the authors should record from PV+ interneurons in wild-type and mutant mice under in vivo conditions (some tetrode recordings might be affordable). At the very least, the results should be better discussed in the context of in vivo activity.

While we agree that examining function of *LIS1* mutant PV+INTs in vivo would be interesting, tetrode recordings would not allow for separation of FS and IS cells. Many exciting possibilities remain for investigating PV+INTs in vivo, however this is not the intention of our study. Furthermore, at this time the McBain lab remains closed until the new year at the earliest. We have been anticipating establishing an in vivo recording setup over the last year but due to vendors being unable to enter campus and the lack of personnel allowed on campus at any one time we have shelved our plans to build an in vivo system until next year.

7) The mechanisms underlying the differences in excitability and synaptic output between fast and intermediate spikers remains unclear. Changes in in excitability are likely to be related to changes in Na^+^ or K^+^ channel density or subtype. Similarly, changes in inhibitory output will be due to changes in q, N, or pR. Both are only tangentially related to the RNAseq data reported in the paper. More work seems needed to work out the mechanisms.

We agree that the mechanisms underlying the differences in excitability and synaptic output between FS and IS cells remains unclear. In regard to the comment that “Changes in in excitability are likely to be related to changes in Na^+^ or K^+^ channel density or subtype”, we observed misrelation of several potassium channel family members (*Kcnj3, Kcnq5, Kcnh7, Kcnb2, Kcnk2, Kcnip1, Kcnip2)* in mutant PV+INTs from the RNAseq experiment (Figure 9). While this does not fully explain the emergence of IS cells, it provides a partial explanation, as subtle alterations to ion channels can alter intrinsic electrical properties. We agree that investigating quantal transmission would contribute to our understanding of the specific changes occurring on that scale, however we did not investigate this in great detail, and many additional experiments are required, that we are presently unable to perform, to fully answer these questions (see Lawrence et al., 2003). We have, however, further analyzed our RNA-seq dataset and identified a small subset of differentially expressed genes which interact with LIS1 and may be interesting targets for future research (this has been added to Figure 7 and to the manuscript).

8) A bit more emphasis on whether there were any indications in the non-autonomous Emx1-cre mutant would have a made a nice complement (might have missed this if it was there).

Analysis of PV+INT morphophysiology in the EmxLis mutant is summarized in Figure 3—figure supplement 1 and Figure 4—figure supplements 2-4.

9) Another concern is with confusing IS cells with the SST populations, as they are both more or less IS and they apparently had no specific marker to identify the PV cells other than the physiology that was sufficiently perturbed to confound proper identification of this population. That they were discernible seemed evident from the scRNA-seq analysis but some attention to the SST interneurons (which clearly would be affected in the nkx2.1-cre KO) would have been a nice addition. Are they still facilitating, is their morphology normal, Are they properly dendritically targeting. Both distinguishing these for the IS population and describing this population at least a bit more thoroughly would have been very nice additions to what is clearly a wonderful paper.

The reviewer may have mistakenly thought that we used the Nkx2.1-Cre for the RNAseq. In the manuscript we state that we used the PV-Cre line for this experiment. As explained in the Materials and methods section, *Sox2*-Cre expression in the mother enables Cre-Lox recombination without Cre expression in the recombined pups (https://www.jax.org/strain/008454). This strategy was used to rederive the Cre-independent GlobalLis (*LIS1*+/-) mice. These mice were then crossed to PV-Cre/Sun1-GFP mice to enable specific targeting of PV+ nuclei. Regarding the potential confusion of IS cells with SST+ cells, we believe this is highly unlikely for several reasons. First, we used PV-TdTomto reporter mice for all recordings, and we demonstrated that there is very high colocalization between PV-TdTomato and PV-IHC and this is unchanged in the GlobalLis mutant. Second, we only observe IS cells following cell-autonomous *LIS1* mutations within interneurons (GlobalLis and NxkLis). If we were simply recording SST+ cells due to faulty TdTomato labeling, we would expect to find these cells in the WT and EmxLis genotypes. Third, IS cells are the dominant PV+INT subtype in the GlobalLis and NkxLis genotypes, if these cells represented SST+ cells, we would not expect to see them represent half of the total PV+ population.

10) The presentation of the manuscript requires improvement. The amount of description of wild type PV interneurons is surprising, as it has already been thoroughly described, in particular in Pelkey et al., 2017. This is necessary for the comparison of wt and mutant cells but perhaps could be parred down rather than spending the first third of the paper describing things are already well documented in the literature. In the present form, the Discussion is rudimentary and focused on side-issues. Several other aspects need to be included (see other major points).

We have worked to improve the clarity of the presentation however we respectfully disagree with the notion that there is too much description of WT PV+INTs in this manuscript, and we disagree that first third of this manuscript was focused on “describing things that are already well documented in the literature”. Pelkey et al., 2017 does not describe PV+ NFS cells nor does it describe PV+ radiatum-targeting cells. These morphophysiological subtypes of PV+INTs are uncharacterized in the literature and we felt that they required adequate description. The Discussion section has also been expanded.